# BioSASANet: Self-Interpretable Shapley Attribution for Deep Genomics Sequence Modeling

## Abstract

Deep learning models have significantly advanced genomics sequence modeling by uncovering complex patterns, but their black-box nature limits the biological insights that can be derived from their predictions. While recent efforts have employed post-hoc methods to identify important nucleotides or motifs, these approaches are decoupled from the prediction process and often suffer from accuracy limitations. In this paper, we introduce the Shapley Additive Self-Attribution (SASA) framework to genomics sequence modeling and propose BioSASANet. BioSASANet integrates a marginal contribution-based sequential module that captures long-range nucleotide interactions, and a positional Shapley value module that explicitly models the reverse complementarity (RC) property of genomic sequences, enabling position-aware, biologically grounded self-attribution. Experiments on two genomics tasks show that BioSASANet achieves accurate predictive performance while providing more faithful nucleotide-level attributions. Additionally, its flexible design allows integration with state-of-the-art DNA language model backbones, enabling advanced models with Shapley value-based self-interpretation.

## 1 INTRODUCTION

The availability of large-scale functional genomics data and advancements in deep learning techniques have brought progress in predicting various genomic properties directly from DNA sequences. Developing models on genomic DNA sequences for the prediction of properties and understanding of transcriptional regulation has long been a central task of computational genomics research. DeepBind Alipanahi et al. (2015) and DeepSEA Zhou & Troyanskaya (2015) were two of the first methods leveraging shallow CNNs for predicting TF binding and chromatin features, respectively. ChromTransfer Salvatore et al. (2023) model chromatin state in a cell type specific manner by predicting the presence of DNase-I peaks. Enformer Avsec et al. (2021a) introduced a transformer-based model that combines convolutional downsampling with long-range self-attention to achieve state-of-the-art genome-wide chromatin feature prediction. Borzoi Linder et al. (2025) builds on Enformer by adding a U-net architecture to restore high-resolution signals, enabling accurate prediction of RNA-seq coverage and competitive variant effect inference. Despite superior predicitive performance, most of these deep learning models are black-boxes providing limited insights into the underlying biological mechanisms. In genomics, however, interpretability is particularly indispensable. Without understanding which nucleotides or motifs drive a model's predictions, researchers cannot verify whether the model captures true biological mechanisms, thus limiting its reliability and usefulness for guiding downstream experimental designs Chen et al. (2024).

The two primary interpretable machine learning (IML) approaches used to explain prediction models in genomics applications are post hoc and self-interpretable explanations Chen et al. (2024). Post hoc explanations are flexible and model agnostic, which are applied after the training of a prediction model. They assign each input feature, such as a nucleotide in a DNA sequence, an importance value based on its contribution to the model prediction. These importance values can be calculated in one of two ways: (1) gradient-based methods (for example, DeepLIFT Shrikumar et al. (2017)) and (2) perturbation-based methods (for example, SHAP/DeepExplainer Lundberg & Lee (2017) and LIME Ribeiro et al. (2016)). Although post hoc algorithms can offer model-agnostic interpretation, the

inherent lack of transparency in models hinders accurate understanding of their internal knowledge. Beyond post hoc explanations, self-interpretable approaches have emerged in computational biology. Exisiting methods construct biologically informed neural networks or incorporate attention mechanisms. Biologically informed networks embed domain knowledge into neural architectures, e.g., DCell Ma et al. (2018), by mapping hidden nodes in the neural network to biological entities or pathways. However, such designs are application-specific and non-generalizable, and the relative importance of the biological entities is usually derived from self-defined scores (e.g., the relative local improvement in predictive power score defined in DCell Ma et al. (2018)) that lack rigorous theoretical backing. Attention mechanisms, widely adopted in transformer-based genomics models (e.g., Enformer Avsec et al. (2021a), Geneformer Theodoris et al. (2023)), provide automatically learned weights that highlight input regions and are considered as an explanation. But their validity and reliability as faithful explanations remain debated Serrano & Smith (2019); Jain & Wallace (2019). These limitations underscore the need for generalizable self-interpretable approaches that offer inherent transparency and rigorous theoretical guarantees of attribution fidelity.

To address the aforementioned challenges, we propose BioSASANet, a biological shapley additive self-interpretable neural network that builds upon the Shapley Additive Self-Attributing Neural Network (SASANet). The network consists of two key components: a **nucleotide marginal contribution based sequential module (NMCSM)** and a **positional shapley value module (PSVM)**. In order to model the long-range nucleotides interaction and reverse complementarity (RC) property of genomics sequence, we utilize Mamba and its bidirectional, reverse complementary equivariant version to model the contextual representations of each nucleotide based on its prefix and the overall sequence context, respectively. To stabilize training and make sure the learned attribution values converge to the output's shapley value, we incorporate an internal distillation and feature-subset label expectation learning strategy. Besides, BioSASANet's flexible architecture allows seamless integration with diverse backbone sequential models accommodating unique characteristics of different biological sequence types, for example, the transformer based and structure state space based DNA language models, demonstrating BioSASANet's broad applicability. Through experiments on two real-world genomics datasets from two different species, we demonstrate that BioSASANet provides more faithful and fine-grained nucleotide-level attributions than popular post-hoc methods. Specifically, BioSASANet accurately localizes histone mark peaks in human ChIP-seq datasets and assign high attribution values to core promoter elements in plant promoter sequences datasets.

Our key contributions are summarized as follows:

- We formulate BioSASANet, the first biological Shapley additive self-interpretable neural network for genomics sequential tasks. It unifies predictive performance with theoretically grounded nucleotide-level attribution while explicitly accounting for key genomic sequence characteristics, including long-range nucleotide interactions and reverse complementarity (RC).

- We demonstrate BioSASANet's adaptability to multiple state-of-the-art biological backbones, such as transformer based and structure state space based DNA language models, highlighting its potential as a generalizable tool for reliable and transparent modeling in computational biology.

- We evaluate BioSASANet on two real-world genomics sequence datasets, demonstrating that our model not only achieves competitive predictive performance but also yields attribution values that faithfully recover biologically meaningful features.

## 2 RELATED WORK

### 2.1 EUKARYOTIC DNA ORGANIZATION AND TERMINOLOGY

Precise regulation of gene expression underlies virtually every biological process in eukaryotes, from development and environmental adaptation to cell differentiation. At the molecular level, DNA is composed of four nucleotide bases: adenine (A), cytosine (C), guanine (G), and thymine (T) which are arranged in two complementary strands that form the iconic double helix through specific base pairing (A with T, and C with G). The genome contains genes—segments of DNA that are transcribed into RNA and can be translated into proteins. Protein-coding genes are structured as introns and exons. For expression, a gene is first transcribed to a pre-mRNA molecule, and introns

are removed via splicing. This combines the exons to one contiguous sequence that encodes the protein. Beyond the genes themselves, the genome harbors regulatory regions—such as promoters, enhancers, silencers, and insulators—that modulate gene expression. These noncoding elements are essential for proper gene regulation. For example, the TATA box, a core promoter element with the consensus sequence TATA(A/T)A(A/T), is recognized by the TATA-binding protein, a subunit of TFIID, and plays an important role in recruiting the basal transcription machinery and in determining the transcription start site (TSS) location. Smale & Kadonaga (2003)

## 2.2 IML in Computational Biology

Recent advances in deep learning for genomics have motivated the use of interpretable machine learning (IML) to ensure that predictive models reflect true biological mechanisms. IML methods largely fall into two categories: post-hoc attribution and self-interpretable modeling. Post-hoc methods assign feature importance after training but do not reveal the model's internal reasoning. Representative examples include BPNet Avsec et al. (2021b), which uses DeepLIFT to derive nucleotide-level contribution scores for TF binding, and Borzoi Linder et al. (2025), which applies Gradient × Input to identify cis-regulatory motifs influencing RNA expression. In contrast, self-interpretable models embed biological domain knowledge directly into the architecture. Examples include DCell Ma et al. (2018), which maps network units to hierarchical cellular subsystems, and P-NET Elmarakeby et al. (2021), which aligns hidden nodes with biological pathways. However,these models are application-specific and their importance measures are heuristic and lack theoretical grounding. Another line of self-interpretable models incorporate and learn a set of weights indicating the amount of attention the model is assigning to specific parts of the input. Attention-based models such as Enformer Avsec et al. (2021a) and Geneformer Theodoris et al. (2023) inspect attention weights to infer regulatory interactions. However, the validity and reliability of such an approach to explain the model's reasoning remains debatable. Unlike these approaches, the proposed BioSASANet provides a generalizable, self-interpretable architecture with theoretical guarantees of attribution fidelity.

# 3 PRELIMINARIES

## 3.1 SASANet: Shapley Additive Self-Attributing Neural Network

BioSASANet's main framework is adapted from SASANet Sun et al. (2025). SASANet consists of two key modules, an intermediate marginal contribution-based sequential modeling module and a positional Shapley value module. Take a DNA sequence $\mathbf{x}$ as an example, for any nucleotide permutation $O \in \pi(\mathcal{N})$ where $\mathcal{N} = \{1, 2, ..., N\}$ is the nucleotides' index set, the marginal contribution of nucleotide appearing at position $O_i$ based on previous nucleotides $\mathbf{x}_{O_{1:i-1}}$ is explicitly modeled as $\triangle(\mathbf{x}_{O_i}, \mathbf{x}_{O_{1:i-1}}; \theta_\triangle)$ by the **marginal contribution-based sequential module**. This allows a permutation-variant output for any nucleotide subsequence $\mathbf{x}_\mathcal{S}$ where $\mathcal{S} \subseteq \mathcal{N}$: $f_c(\mathbf{x}_\mathcal{S}, O_\mathcal{S}; \theta_\triangle) = \sum_{i=1}^{|\mathcal{S}|} \triangle(\mathbf{x}_{O_{\mathcal{S}i}}, \mathbf{x}_{O_{\mathcal{S}1:i-1}}; \theta_\triangle) + \phi_0$, capturing the intermediate effects of each given nucleotide accounting for previous nucleotides. $\phi_0$ represents the model output when no features are observed. The **positional Shapley value module** $\phi(\mathbf{x}_\mathcal{S}; \theta_\phi)_{i,k}$ is trained to estimate the contribution of feature $i$ at position $k$ within sample $\mathbf{x}_\mathcal{S}$ to the intermediate output $f_c$. It is a permutation-invariant network that models attribution values for individual features in samples of arbitrary size.

A distillation loss was constructed for each feature $i \in \mathcal{S}$ in variable-size input $\mathbf{x}_\mathcal{S} \in \bigcup_{k=1}^N \mathbb{R}^k$ as in eq 1, where $\mathcal{D} \subset \pi(\mathcal{S})$ is permutations drawn for training.:

$$L_s^{(i,k)}(\mathbf{x}_\mathcal{S}) = \frac{1}{|\mathcal{D}|} \sum_{O \in \mathcal{D}} \mathbb{I}\{O_k = i\}(\phi(\mathbf{x}_\mathcal{S}; \theta_\phi)_{i,k} - \triangle(\mathbf{x}_i, \mathbf{x}_{O_{1:k-1}}; \theta_\triangle))^2. \tag{1}$$

Given a set of training samples $\mathcal{D}_{tr}$, a loss for a feature subset $\mathbf{x}_\mathcal{S}$ under input permutation $O_\mathcal{S}$ was defined as

$$L_v(\mathbf{x}_\mathcal{S}, O_\mathcal{S}) = \frac{\sum_{(\mathbf{x}', y') \in \mathcal{D}_{tr}} \mathbb{I}\{\mathbf{x}'_\mathcal{S} = \mathbf{x}_\mathcal{S}\} L_m(\mathbf{x}_\mathcal{S}, y', O_\mathcal{S})}{\sum_{(\mathbf{x}', y') \in \mathcal{D}_{tr}} \mathbb{I}\{\mathbf{x}'_\mathcal{S} = \mathbf{x}_\mathcal{S}\}}. \tag{2}$$

where $L_m$ is the loss function based on the intermediate output $f_c$ depending on the training task.

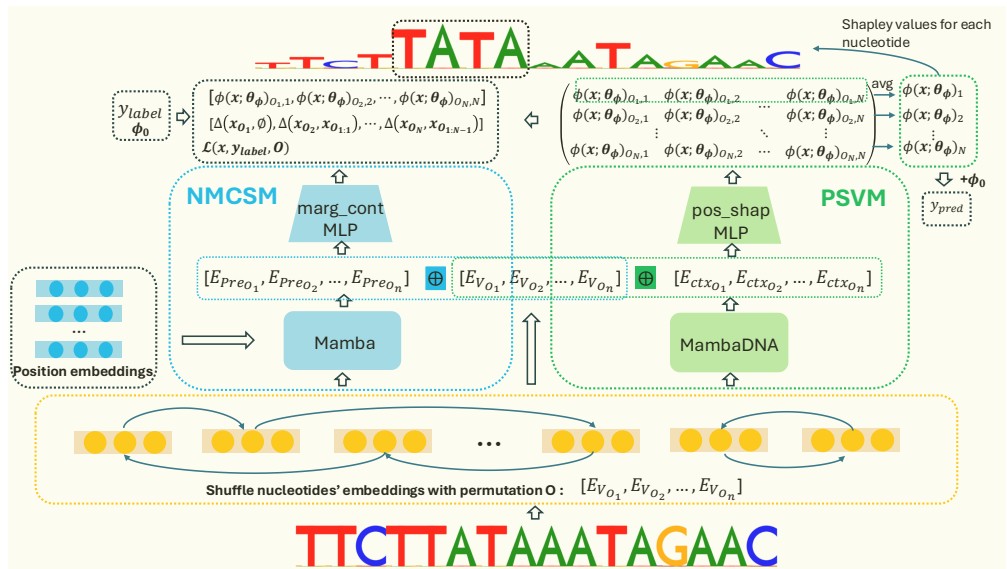

Figure 1: Overview of the BioSASANet framework.

The network will learn to generalize by understanding permutation patterns from sampled permutations during training, without the need to cover all possible permutations exhaustively. The details of the proof can be found in the paper Sun et al. (2025).

**Theorem 3.1.** *Optimizing $L_s^{(i,k)}(\mathbf{x}_{\mathcal{S}})$ and $L_v(\mathbf{x}_{\mathcal{S}}, O_{\mathcal{S}})$ with $\mathcal{D}_{tr}$ for sampled permutations $O_{\mathcal{S}} \subset \pi(\mathcal{S})$ makes $\phi$ converge to Shapley value of the final output.*

# 4 BIOSASANET METHODS

## 4.1 FRAMEWORK OVERVIEW

The structure overview of BioSASANet can be found in figure 1. We firstly embed each position's nucleotide with a unique embedding table having four values representing the four nucleotides: A,T,G,C. In the **nucleotide marginal contribution-based sequential modeling** module, we utilize Mamba Gu & Dao (2023) to obtain each position's prefix features' representation considering its long-range sequential dependency modeling ability with linear complexity in sequence length. To help identify sequential patterns and facilitate the convergence of the model, we also incorporate position embeddings for each nucleotide location within the permutation. Concatenating each nucleotide's embedding and its corresponding prefix embedding, we model the marginal contribution of each nucleotide appearing at position $O_i$: $\triangle(\mathbf{x}_{O_i}, \mathbf{x}_{O_{1:i-1}}; \theta_\triangle)$ with a feed-forward network. The **positional Shapley value module** models the attribution value of each feature in a given sample. Similar to the marginal contribution module, we firstly adapt the MambaDNA block introduced in Caduceus Schiff et al. (2024) to get the overall sample representation for a specific feature. Then we concatenate this with each corresponding nucleotide's embedding and model the positional Shapley value $\phi(\mathbf{x}_{\mathcal{S}}; \theta_\phi)_{i,k}$ via another feed-forward neural network for nucleutide i appearing at position k. We exclude position embeddings to maintain the permutation invariance property asked for the convergence of $\phi$ to the Shapley value in $f(\mathbf{x}; \theta_\phi)$. Details for each module are shown below.

## 4.2 **NMCSM**: NUCLEOTIDE MARGINAL CONTRIBUTION-BASED SEQUENTIAL MODELING

We utilize Mamba structure as part of our MCSM to get each position's prefix nucleotides' representation. We first review the foundation of Structured State Space Models (SSMs), highlighting their classical linear time-invariant (LTI) formulation. We then describe the selective SSM variant introduced in Mamba. Finally, we outline the Mamba block, which integrates the selective SSM with a gated MLP mechanism to effectively capture long-range dependencies while maintaining efficient sequence processing.

**Structured State Space Models** Structured state space sequence models (S4) are a recent class of sequence models for deep learning that are broadly related to RNNs, CNNs, and classical state space models. They are inspired by a particular continuous system that maps a 1-dimensional sequence $\mathbf{x} \in \mathbb{R}^T \mapsto \mathbf{y} \in \mathbb{R}^T$ through an implicit latent state $\mathbf{h} \in \mathbb{R}^{(N,T)}$. A general discrete form of structured SSMs takes the form of equation 3:

$$\mathbf{h}_t = \mathbf{A}\mathbf{h}_{t-1} + \mathbf{B}\mathbf{x}_t \quad (3a) \qquad\qquad \mathbf{h}_t = \mathbf{A}_t\mathbf{h}_{t-1} + \mathbf{B}_t\mathbf{x}_t \quad (4a)$$

$$y_t = \mathbf{C}^\top \mathbf{h}_t \quad (3b) \qquad\qquad y_t = \mathbf{C}_t^\top \mathbf{h}_t \quad (4b)$$

where $\mathbf{A} \in \mathbb{R}^{(N,N)}, \mathbf{B} \in \mathbb{R}^{(N,1)}, \mathbf{C} \in \mathbb{R}^{(N,1)}$. Structured SSMs are so named because the $\mathbf{A}$ matrix controlling the temporal dynamics must be structured in order to compute this sequence-to-sequence transformation efficiently enough to be used in deep neural networks. The original structures introduced were diagonal plus low-rank (DPLR)Gu et al. (2021) and diagonalGu et al. (2022), which remains the most popular structure.

**Selection Mechanism** The form 4 where the parameters $\mathbf{A}$, $\mathbf{B}$, $\mathbf{C}$ can also vary in time was introduced in Mamba as the selective SSM. Compared to the standard LTI formulation 3, this model can selectively choose to focus on or ignore inputs at every timestep. It was shown to perform much better than LTI SSMs on information-dense data such as language, especially as its state size N increases allowing for more information capacity.

**Mamba** The Mamba block presented in Gu & Dao (2023) is formed by combining a selective SSM sequence transformation and a gated MLP mechanism. An incoming sequence is copied and projected to twice the input dimension. One copy is then passed through a causal convolution, followed by the SiLU/Swish activation Ramachandran et al. (2017) and then finally through the selective SSM. The other copy has the SiLU non-linearity applied to it and then gates the SSM output. The gated representation is then projected back to the original dimension. This is a causal, left-to-right sequence operation, which aligns perfectly with our goal of modeling each position's prefix nucleotides' representation.

## 4.3 PSVM: POSITIONAL SHAPLEY VALUE MODULE

The Positional Shapley value module models the attribution value of nucleotide $i$ appearing at position $k$ with respect to current prediction for gene sequence $\mathbf{x}_\mathcal{S}$: $\phi(\mathbf{x}_\mathcal{S}; \theta_\phi)_{i,k}$. Relying solely on a nucleotide's own value embedding is far from sufficient to accurately model its attribution to the overall prediction. To get more expressive representation for each nucleotide in a gene sequence, there are several challenges such as long-range nucleotide interaction modeling Nguyen et al. (2023), the effects of both upstream and downstream regions of the genome and the reverse complementary(RC) property of DNA Schiff et al. (2024). Therefore, we need to figure out a way to model the representation for each specific nucleotide considering these challenges to better model each nucleotide's attribution for the final output. We utilize MambaDNA block introduced in Schiff et al. (2024) which is a bi-directional and RC-equivariant extension of Mamba Gu & Dao (2023). Details of MambaDNA are shown below. We firstly introduce BiMamba, which is the core component of MambaDNA.

**BiMamba** The standard Mamba block is a causal, left-to-right sequence operation. To capture both upstream and downstream nucleotides' effects, Schiff et al. (2024) convert it to bi-directional Mamba called BiMamba. They achieve this by applying the Mamba block twice: once to the original sequence and once to a copy that is reversed along the length dimension. To combine the information from both directions, the output of the reversed sequence is flipped along the length dimension again and added with the output of the original sequence. To avoid the doubled memory cost, they implement the BiMamba block with shared projection weights between the forward and backward Mamba, which accounts for a vast majority of the model's parameters compared to those in the convolution and the SSM submodules.

**MambaDNA: Reverse complementary equivariant Mamba** To incorporate the RC equivariance inductive bias into the module design, Schiff et al. (2024) apply a BiMamba block on a sequence

and its RC with parameters shared between the two applications. Concretely, they split a sequence of length T with D channels along the channel dimension into two halves:

$$split(\mathbf{X}_{1:T}^{1:D}) := [\mathbf{X}_{1:T}^{1:(D/2)}, \mathbf{X}_{1:T}^{(D/2):D}] \tag{5}$$

The first half is sent to the BiMamba block, while the second half receives the RC operation before sending to the same BiMamba block. The RC operation is defined as:

$$RC(\mathbf{X}_{1:T}^{1:D}) := \mathbf{X}_{T:1}^{D:1} \tag{6}$$

Finally, the output of the first half and the RC of the output of the second half are re-concatenated along the channel dimension together to form the final output. Let $M_\theta$ denotes the BiMamba block, the whole pipeline of RC equivariant Mamba block MambaDNA $M_{RC_e,\theta}$ can be expressed as follows:

$$M_{RC_e,\theta}(\mathbf{X}_{1:T}^{1:D}) := concat([M_\theta(\mathbf{X}_{1:T}^{1:(D/2)}), RC(M_\theta(RC(\mathbf{X}_{1:T}^{(D/2):D})))]) \tag{7}$$

They prove in Schiff et al. (2024) that MambaDNA satisfies the RC equivariance property desired for processing DNA sequences, i.e., $\mathrm{RC} \circ \mathbf{M}_{\mathrm{RCe},\theta}\left(\mathbf{X}_{1:T}^{1:D}\right) = \mathbf{M}_{\mathrm{RCe},\theta} \circ \mathrm{RC}\left(\mathbf{X}_{1:T}^{1:D}\right)$.

In this way, MambaDNA can perfectly solve the above mentioned 3 challenges to get expressive representation for each nucleotide in a gene sequence. The inherent Mamba structure enables effective long-range nucleotide interaction modeling with linear complexity in sequence length. The bi-directional design of BiMamba captures the effects of both upstream and downstream regions of the gene sequence. The RC property of DNA are incorporated into the MambaDNA module design. Concatenating the expressive representation with each nucleotide's own value embedding, we utilize another feed-forward neural network to model the positional Shapley value for each nucleotide i appearing at position k in gene sequence $\mathbf{x}_S$: $\phi(\mathbf{x}_S; \theta_\phi)_{i,k}$.

## 4.4 TRAINING LOSS

The final loss for a sample $(\mathbf{x}, y)$ with permutation $O$ consists of two parts: the positional Shapley value distillation loss and the feature subset label expectation loss. We denote $\lambda_s$ and $\lambda_v$ as the hyperparameters to balance the two losses. The final loss for regression task is shown in equation 8. For classification tasks, the feature subset label expectation loss (the second term in Equation 8) can be adapted accordingly.

$$L(\mathbf{x}, y, O) = \sum_{k=1}^{N} (\lambda_s(\phi(\mathbf{x}; \theta_\phi)_{O_k,k} - \triangle(\mathbf{x}_{O_k}, \mathbf{x}_{O_{1:k-1}}; \theta_\triangle))^2 + \lambda_v(\sum_{i=1}^{k} \triangle(\mathbf{x}_{O_i}, \mathbf{x}_{O_{1:i-1}}; \theta_\triangle) + \phi_0 - y)^2). \tag{8}$$

## 5 EXPERIMENTS

### 5.1 DATASETS

#### 5.1.1 EPIGENOMIC HISTONE MODIFICATION DATASET

We used ChIP-seq data for four histone marks (H3K4me3, H3K27me3, H3K9me3, H3K36me3) in the K562 human cell line from the ENCODE project Consortium et al. (2012), using their corresponding BED narrowPeak files. Following Dalla-Torre et al. Dalla-Torre et al. (2025), we constructed binary classification datasets by selecting 1-kb sequences overlapping peaks as positives and those without peaks as negatives. These datasets were used to train our framework and assess nucleotide-level attributions against the ground-truth peak intervals.

#### 5.1.2 PLANT CORE PROMOTER DATASET

The plant core promoter datasets consist of 170-bp sequences ($-165$ to $+5$ relative to TSS) from Arabidopsis thaliana, Zea mays, and Sorghum bicolor. Promoter strength was measured using

STARR-seq in two assay systems: tobacco leaves (Plant Leaf) and maize protoplasts (Plant Proto), normalized to the viral 35S minimal promoter with values averaged over two replicates. These datasets were used to train our framework to predict promoter strength and uncover core promoter elements (CPEs), such as the TATA box and Y Patch motifs. Following Jores et al. Jores et al. (2021), CPEs were localized by scanning sequences with PWMs from the original MEME files, retaining motif hits with similarity scores above 0.85 as ground-truth regions for attribution evaluation.

## 5.2 EXPERIMENTAL SETUP

### 5.2.1 IMPLEMENTATION DETAILS

The details of main model parameters setting can be found in table 3 in appendix A.4. We used the Adam optimizer Kingma (2014) to optimize the model parameters. We utilize the early stopping learning strategy with patience set to 10 and 15 for the epigenomic histone modification dataset and the plant core promoter dataset respectively. All experiments were conducted on a single NVIDIA RTX A6000 GPU with CUDA version 12.2. The pseudocode of BioSASANet's training and inference procedure can be found in 1 and 2 in appendix A.5.

### 5.2.2 COMPARED METHODS AND EVALUATION METRICS

We compared our self-interpretable neural network BioSASANet with other three popular post hoc explanation methods, covering both gradient-based and perturbation-based types: DeepLift Shrikumar et al. (2017), GradientSHAP Lundberg & Lee (2017), and Lime Ribeiro et al. (2016). We used the Captum library Kokhlikyan et al. (2020) to implement these methods.

To evaluate the faithfulness of the nucleotide-level attributions produced by BioSASANet, we adopt the mean precision at top-k (MP@k) metric. For each input sequence, we first rank all nucleotide positions based on their attribution scores in descending order. We then calculate the proportion of the top-k positions that fall within the ground-truth annotated functional regions (e.g., known transcription factor binding sites or core promoter motifs such as the TATA-box). The final MP@k score is obtained by averaging this precision across all test samples. A higher MP@k indicates better alignment between the model's high-attribution positions and biologically meaningful sequence elements, thus reflecting more faithful explanations.

### 5.2.3 OTHER BACKBONE DNA LM

To prove BioSASANet's compatibility with different backbones for genomics sequential modeling, we implement two variants of BioSASANet using different state-of-the-art backbone DNA LMs: **Evo2** Brixi et al. (2025) and **HyenaDNA** Nguyen et al. (2023). We compare their performance with the original BioSASANet on the two genomics dataset.

**Evo2** Evo2 Brixi et al. (2025) is a transformer-based DNA language model tailored for genomic sequences. The model includes three main components: (1) Pairwise Bias Computation, which captures inter-nucleotide interactions; (2) MSAColumnAttention, adapted to operate without multiple sequence alignments by using a learned token for single-sequence representation; and (3) Evoformer2Blocks, a series of attention and transition layers adapted to DNA context. Due to its non-causal modeling structure, we choose Evo2-7b as our optional backbone for positional Shapley value module to get each nucleotide's representation based on the overall sequence context. As analysed in Brixi et al. (2025), intermediate embeddings work better than final embeddings on some downstream tasks. Followed Brixi et al. (2025), we extract the embeddings from block 27 of the Evo2-7b model due to their superior performance on the BRCA1 variant supervised classification task.

**HyenaDNA** HyenaDNA Nguyen et al. (2023) is a long-range genomic modeling architecture based on the Hyena operator, a structured state space model that replaces traditional attention mechanisms with subquadratic convolutions. It introduces Hyena filters to efficiently capture long-range nucleotide dependencies while significantly reducing memory and compute cost compared to Transformer-based architectures. We utilize HyenaDNA_large_1m_seqlen as an alternative backbone for the NMCSM module in BioSASANet to model the representations of each nucleotide based on its prefix context considering its causal nature.

Table 1: Interpretability Quantification on histone-mark datasets.

| Eval metrics | Datasets | BioSASANet | DeepLift | GradientSHAP | Lime |
|---|---|---|---|---|---|
| MP@50 | H3K4me3 | **0.7473** | 0.7399 | 0.7145 | 0.6706 |
| | H3K27me3 | **0.7131** | 0.6895 | 0.6527 | 0.5740 |
| | H3K9me3 | **0.5818** | 0.5345 | 0.5324 | 0.4153 |
| | H3K36me3 | **0.8662** | 0.7934 | 0.7015 | 0.4360 |
| MP@100 | H3K4me3 | **0.7302** | 0.7243 | 0.6975 | 0.7065 |
| | H3K27me3 | **0.6878** | 0.6471 | 0.6179 | 0.6323 |
| | H3K9me3 | **0.5233** | 0.4945 | 0.4955 | 0.4815 |
| | H3K36me3 | **0.7614** | 0.6929 | 0.6287 | 0.4744 |
| MP@200 | H3K4me3 | **0.7051** | 0.7001 | 0.6846 | 0.6692 |
| | H3K27me3 | **0.6191** | 0.6002 | 0.5822 | 0.5616 |
| | H3K9me3 | **0.4796** | 0.4594 | 0.4518 | 0.4036 |
| | H3K36me3 | **0.6313** | 0.5900 | 0.5555 | 0.4217 |

## 5.3 RESULTS ANALYSIS

### 5.3.1 EPIGENOMIC HISTONE MODIFICATION DATASET

**Interpretability Quantification** To quantitatively evaluate the interpretability performance of BioSASANet, we conducted experiments on four epigenomic histone modification datasets: H3K4me3, H3K27me3, H3K9me3, and H3K36me3; using mean precision at k (MP@k) as the evaluation metric, with k equals to 50, 100 and 200. We compared BioSASANet against several widely used post-hoc attribution methods, including DeepLIFT, GradientSHAP, and LIME. As shown in Table 1, BioSASANet consistently outperforms all compared post hoc methods across all datasets and evaluation thresholds, with particularly notable gains on H3K36me3 (0.8662 vs. 0.7934 for DeepLIFT and 0.7015 for GradientSHAP) and H3K9me3 (0.5818 vs. 0.5345 for DeepLIFT). These trends persist across MP@100 and MP@200, demonstrating the robustness and reliability of the learned attributions.

**Different Backbone DNA LM Performance Comparison** To evaluate the flexibility and generalizability of BioSASANet across different backbones, we compare BioSASANet with two other variants using Evo2-7b and HyenaDNA for genomics sequential modeling module. Their predictive and interpretability performances on four histone-mark datasets are reported in Table 4 in appendix A.6, using average precision (AP), area under the ROC curve (AUC), and mean precision at top-k (MP@k) as metrics. As for the predictive performance evaluation (by AP and AUC), all three frameworks achieve competitive predictive performance across datasets, with slight variations across tasks. Notably, BioSASANet shows superior AP and AUC on H3K4me3 and H3K36me3, while HyenaDNA excels on H3K27me3 and H3K9me3. In terms of interpretability, BioSASANet consistently outperforms the other two variants across most histone marks datasets on all MP@k levels. The other two BioSASANet variants also demonstrate strong performance under specific settings. For instance, Evo2-7b achieves the highest MP@50 on H3K27me3 and H3K9me3, while HyenaDNA attains the best MP@200 on H3K9me3.

**Case Study: BioSASANet Accurately Captures the Peak Regions of Histone Marks** To qualitatively evaluate the interpretability of BioSASANet, we visualize the attribution maps generated on representative samples from each histone-mark dataset. As shown in Figure 2 shades indicate higher contributions to the prediction. The blue triangles mark the top-50 nucleotide positions with the highest attribution scores, while the yellow-highlighted regions correspond to the ground-truth peak intervals of histone modifications. Across all four histone marks—H3K4me3, H3K27me3, H3K9me3, and H3K36me3, BioSASANet successfully identifies the biologically meaningful peak regions. This case study visually demonstrates that BioSASANet not only provides accurate predictions but also produces faithful and fine-grained nucleotide-level attributions that reflect true biological signal, further confirming the interpretability strength of the model.

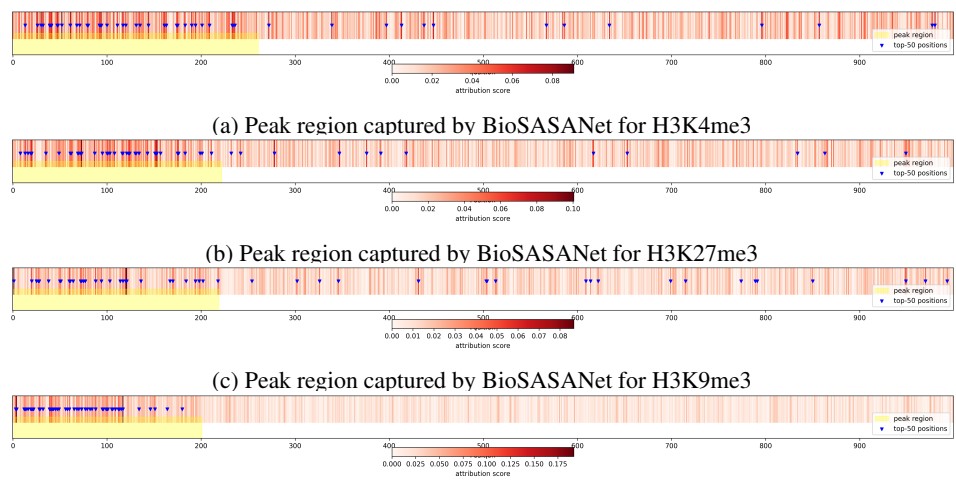

(a) Peak region captured by BioSASANet for H3K4me3

(b) Peak region captured by BioSASANet for H3K27me3

(c) Peak region captured by BioSASANet for H3K9me3

(d) Peak region captured by BioSASANet for H3K36me3

Figure 2: Peak regions captured by BioSASANet for four histone marks.

Table 2: Interpretability Quantification on plant datasets.

| Eval metrics | Datasets | BioSASANet | DeepLift | GradientSHAP | Lime |
|---|---|---|---|---|---|
| MP@5 | Plant Leaf | **0.1385** | 0.1266 | 0.1289 | 0.1242 |
| | Plant Proto | **0.1489** | 0.1411 | 0.1401 | 0.1254 |
| MP@10 | Plant Leaf | **0.1450** | 0.1273 | 0.1358 | 0.1216 |
| | Plant Proto | **0.1487** | 0.1401 | 0.1406 | 0.1171 |
| MP@15 | Plant Leaf | **0.1413** | 0.1247 | 0.1335 | 0.1177 |
| | Plant Proto | **0.1463** | 0.1376 | 0.1420 | 0.1134 |

**Parameter Experiments** To investigate the impact of key hyperparameters on the interpretability of BioSASANet, we conducted controlled experiments on the H3K36me3 dataset, evaluating mean precision at top- (MP@k) with k equals to 50, 100 and 200. As shown from Figure 4d to 4f in appendix A.7, we explore the effects of three parameters: the nucleotide embedding size, the hidden dimension of the marginal contribution module and the hidden dimension of the Caduceus-based positional shapley value module. Through the experiments' results, we can select the reasonable parameter settings to get the best model interpretability performance.

### 5.3.2 PLANT CORE PROMOTER DATASET

**Interpretability Quantification**

We compare BioSASANet with post-hoc explanation methods on two plant datasets (Leaf and Proto) with the same experiment setting as the histone mark datsets. As shown in Table 2, BioSASANet consistently outperforms the three post-hoc methods across all $k$ values, demonstrating its superior ability to localize key nucleotides. Similar trends are observed on the Leaf dataset, highlighting the robustness of our self-attributing framework.

**Different Backbone DNA LM Performance Comparison**

We further investigate the impact of replacing the backbone sequence encoder in BioSASANet with two state-of-the-art DNA language models: Evo2-7b, and HyenaDNA as in histone marks dataset. As shown in Table 5 in appendix A.6, Evo2-7b achieves the best interpretability performance on the Plant Leaf dataset, attaining the highest MP@5 (0.1762), MP@10 (0.1747), and MP@15 (0.1723), significantly outperforming the other variants. On the Plant Proto dataset, the original BioSASANet yields the lowest MSE (1.3017) and the best interpretability metrics across the board, indicating

better attribution fidelity. These results demonstrate that BioSASANet can flexibly benefit from stronger backbone representations to boost both predictive and explanatory performance.

**Case Study: BioSASANet Accurately Identify the Core Promoter Regions**

We conduct a case study to evaluate whether the attributions generated by BioSASANet can correctly highlight known regulatory elements within plant promoter sequences. As shown in figure 3, for each dataset, we select three representative promoter sequences and visualize the base-level attribution scores along the full 170-bp input window. In each plot, the height of each nucleotide reflects its contribution to the predicted promoter strength—taller letters indicate higher attribution. At the top of each panel, we display canonical position-weight matrices (PWMs) for major core promoter elements used as references. The black rectangles highlight the experimentally validated locations of core promoter elements within each sequence (ground truth). BioSASANet produces concentrated high-attribution signals that align precisely with known core promoter elements such as the TATA box and Y-patch, demonstrating that the model not only predicts promoter strength accurately, but also assigns biologically faithful importance to the correct regulatory loci. Jores et al. (2021). Due to page limitations in the main text, we have placed the case study on plant leaf dataset in figure 5 in A.8 part of appendix.

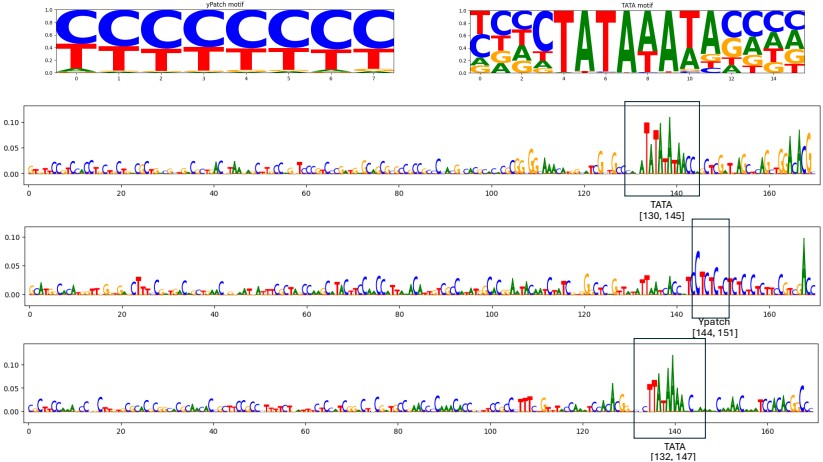

Figure 3: Motif discovery from Shapley values (Proto).

**Parameter Experiments**

We conducted parameter sensitivity analysis on the two plant dataset, varying three key hyperparameters same as the histone mark datasets. As shown in Figure 4a to 4c in appendix A.7, all MP@$k$ metrics consistently peak when the embedding size is set to 128, outperforming both smaller (64) and larger (256) dimensions. For the hidden dimension of the marginal contribution module, performance steadily improves with increased capacity, reaching the best results at 128. A similar upward trend is observed for the hidden dimension of the Caduceus-based positional shapley value module, where the highest MP@$k$ scores are again achieved at 128. These results suggest that richer representations in both the NMCSM and positional shapley value modules enhance interpretability.

## 6 CONCLUSION

We introduce BioSASANet, a novel self-interpretable neural network that unifies prediction and attribution through a Shapley Additive Self-Attention framework. By explicitly modeling marginal contributions and positional shapley values over nucleotides, BioSASANet achieves strong performance on both prediction accuracy and interpretability. Experiments on two genomics datasets show that BioSASANet achieves comparable predictive performance while providing more faithful nucleotide-level attributions.

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

## A APPENDIX

### A.1 ETHICS STATEMENT

This work adheres to the ICLR Code of Ethics. In this study, no human subjects or animal experimentation was involved. All datasets used, including the epigenomic histone modification dataset and the plant core promoter dataset, were sourced in compliance with relevant usage guidelines, ensuring no violation of privacy. We have taken care to avoid any biases or discriminatory outcomes in our research process. No personally identifiable information was used, and no experiments were conducted that could raise privacy or security concerns. We are committed to maintaining transparency and integrity throughout the research process.

## A.2 REPRODUCIBILITY STATEMENT

We have made every effort to ensure that the results presented in this paper are reproducible. All code and datasets will make publicly available to facilitate replication and verification when the paper is published. The experimental setup, including training steps, model configurations, and hardware details, is described in detail in the paper.

Additionally, the used datasets: including the epigenomic histone modification dataset and the plant core promoter dataset, are publicly available, ensuring consistent and reproducible evaluation results.

We believe these measures will enable other researchers to reproduce our work and further advance the field.

## A.3 LLM USAGE

Large Language Models (LLMs) were used to aid in the writing and polishing of the manuscript. Specifically, we used an LLM to assist in refining the language, improving readability, and ensuring clarity in various sections of the paper. The model helped with tasks such as sentence rephrasing, grammar checking, and enhancing the overall flow of the text.

It is important to note that the LLM was not involved in the ideation, research methodology, or experimental design. All research concepts, ideas, and analyses were developed and conducted by the authors. The contributions of the LLM were solely focused on improving the linguistic quality of the paper, with no involvement in the scientific content or data analysis.

We have ensured that the LLM-generated text adheres to ethical guidelines and does not contribute to plagiarism or scientific misconduct.

## A.4 MODEL PARAMETERS SETTING

BioSASANet consists of two components: the Nucleotide Marginal Contribution-based Sequential Modeling (NMCSM) module and the Positional Shapley Value Module (PSVM). Table 3 summarizes the key model parameters, hyperparameters, and training configurations used for the two modules across the histone-mark and plant-promoter datasets.

Table 3: Model parameters for two datasets

| Parameter | Histone Marks Datasets | Plant Datasets |
|---|---|---|
| nucleotide_emb_size | 128 | 128 |
| d_state_marginal | 32 | 128 |
| d_conv_marginal | 4 | 4 |
| expand_marginal | 2 | 2 |
| deep_marginal_emb | [128, 128, 128] | [256, 128, 64] |
| n_layer_caduceus | 2 | 4 |
| d_state_caduceus | 32 | 128 |
| d_conv_caduceus | 4 | 4 |
| expand_caduceus | 2 | 2 |
| deep_pos_shap_emb | [128, 128, 128] | [256, 128, 64] |
| $\lambda_v$ | 1.0 | 1.0 |
| $\lambda_s$ | 1.0 | 1.0 |
| lr | 1e-4 | 1e-3 |
| batch_size | 30 | 128 |

## A.5 PSEUDOCODE OF BIOSASANET'S TRAINING AND INFERENCE PROCEDURE

The pseudocodes for the training and inference procedures of BioSASANet are provided in Algorithm 1 and Algorithm 2 respectively. Note that the loss function in line 11 of Algorithm 1 is written for a regression setting. For other task types, such as binary or multi-class classification, the $L_v$ component should be replaced with the corresponding classification loss.

---

**Algorithm 1** Training of BioSASANet

---

**Require:** Genomics sequence length $N$, learning rate $\eta$, number of training rounds $N_{\text{round}}$, batch size $B$, training data $\mathcal{D}_{\text{train}}$, regularization parameters $\lambda_s$, $\lambda_v$, nucleoties' index set $\mathcal{N} = \{1, 2, ..., N\}$
**Ensure:** Model parameters $\theta_\Delta$, $\theta_\phi$, $\phi_0$

1: Initialize $\theta_\Delta$, $\theta_\phi$, $\phi_0$
2: $\Delta_{\theta_\Delta} \leftarrow 0, \Delta_{\theta_\phi} \leftarrow 0, \Delta_{\phi_0} \leftarrow 0$
3: **for** round = 1 to $N_{\text{round}}$ **do**
4:     **for** each $(\mathbf{x}, y) \in \mathcal{D}_{\text{train}}$ **do**
5:         Sample a random order $O \in \pi(\mathcal{N})$
6:         $[\Delta(\mathbf{x}_{O_1}, \emptyset), \Delta(\mathbf{x}_{O_2}, \mathbf{x}_{O_{1:1}}), \dots] \leftarrow \textbf{NMCSM}(\mathbf{x}, O)$
7:         contributions $\leftarrow [0, \Delta(\mathbf{x}_{O_1}, \emptyset), \Delta(\mathbf{x}_{O_2}, \mathbf{x}_{O_{1:1}}), \dots]$
8:         cumulative_values $\leftarrow$ cumsum(contributions) $+ \phi_0$
9:         $pos\_shapley\_matrix \leftarrow \textbf{ShapleyNetwork}(x_{O_{1:N}})$
10:       Extract diagonal: $pos\_shap \leftarrow \text{diag}(pos\_shapley\_matrix)$
11:       $loss \leftarrow \lambda_s \| pos\_shap - \text{contributions} \|_2^2 + \lambda_v \| \text{cumulative\_values} - y \|_2^2$
12:       Accumulate gradients:
13:         $\Delta_{\theta_\Delta} \leftarrow \Delta_{\theta_\Delta} + \frac{\partial loss}{\partial \theta_\Delta}, \quad \Delta_{\theta_\phi} \leftarrow \Delta_{\theta_\phi} + \frac{\partial loss}{\partial \theta_\phi}, \quad \Delta_{\phi_0} \leftarrow \Delta_{\phi_0} + \frac{\partial loss}{\partial \phi_0}$
14:       **if** round % B == 0 **then**
15:         Update parameters:
16:           $\theta_\Delta \leftarrow \theta_\Delta - \eta \Delta_{\theta_\Delta}, \quad \theta_\phi \leftarrow \theta_\phi - \eta \Delta_{\theta_\phi}, \quad \phi_0 \leftarrow \phi_0 - \eta \Delta_{\phi_0}$
17:         Reset accumulations: $\Delta_{\theta_\Delta} \leftarrow 0, \Delta_{\theta_\phi} \leftarrow 0, \Delta_{\phi_0} \leftarrow 0$
18:       **end if**
19:     **end for**
20: **end for**
21: **return** $\theta_\Delta$, $\theta_\phi$, $\phi_0$

---

**Algorithm 2** Inference of BioSASANet

---

**Require:** Genomics sequence $\mathbf{x}$
**Ensure:** Prediction $y_{\text{pred}}$, attribution value $shapley\_values$

1: $pos\_shapley\_matrix \leftarrow \textbf{ShapleyNetwork}(\mathbf{x})$
2: $shapley\_values \leftarrow \text{mean}(pos\_shapley\_matrix, \text{axis} = 1)$
3: $y_{\text{pred}} \leftarrow \text{sum}(shapley\_values) + \phi_0$
4: **return** $y_{\text{pred}}$, $shapley\_values$

---

## A.6   Performance of BioSASANet Variants

To assess the flexibility of BioSASANet across different genomic backbone architectures, we additionally instantiate the framework with Evo2-7b Brixi et al. (2025) and HyenaDNA Nguyen et al. (2023) as substitutes for genomics modeling. Their predictive (AP, AUC) and interpretability (MP@k) performances on the four histone-mark datasets and two plant promoter datasets are reported in Table 4 and Table 5 respectively. Overall, the prediction accuracy and interpretability performance vary across datasets and backbones, demonstrating that the proposed attribution mechanism is broadly compatible with diverse backbone architectures.

## A.7   Parameter Experiments

To assess the sensitivity of BioSASANet to key model parameters, we conduct controlled experiments on both histone-mark (H3K36me3) and plant promoter datasets. Across all datasets, we vary three critical parameters related to the two core modules—(1) nucleotide embedding size, (2) hidden dimension of the marginal-contribution module (NMCSM), and (3) hidden dimension of the Caduceus-based positional Shapley value module (PSVM). As shown in Appendix Fig. 4a to 4f, MP@k metrics (k = 50, 100, 200 for histone mark datasets and k = 5, 10, 15 for plant promoter datasets) consistently indicate that an nucleotide embedding size of 128 and reasonable state hidden dimensions (128 for plant promoter datasets and 32 for histone mark datasets) yield the most infor-

Table 4: Performance of BioSASANet variants on histone-mark datasets.

| Eval metrics | Datasets | BioSASA | BioSASA w Evo2-7b | BioSASA w HyenaDNA |
|---|---|---|---|---|
| AP | H3K4me3 | 0.8230 | **0.8249** | 0.8173 |
| | H3K27me3 | 0.7122 | 0.7177 | **0.7426** |
| | H3K9me3 | 0.5569 | 0.5373 | **0.5924** |
| | H3K36me3 | **0.7103** | 0.6789 | 0.6765 |
| AUC | H3K4me3 | 0.8070 | **0.8088** | 0.7995 |
| | H3K27me3 | 0.7772 | 0.7797 | **0.7804** |
| | H3K9me3 | 0.5752 | 0.5653 | **0.6039** |
| | H3K36me3 | **0.7729** | 0.7335 | 0.7224 |
| MP@50 | H3K4me3 | **0.7473** | 0.7046 | 0.6749 |
| | H3K27me3 | 0.7131 | **0.7244** | 0.5388 |
| | H3K9me3 | 0.5818 | **0.6030** | 0.5261 |
| | H3K36me3 | **0.8662** | 0.6628 | 0.5517 |
| MP@100 | H3K4me3 | **0.7302** | 0.6610 | 0.6605 |
| | H3K27me3 | **0.6878** | 0.6659 | 0.5281 |
| | H3K9me3 | **0.5233** | 0.5176 | 0.5157 |
| | H3K36me3 | **0.7614** | 0.6145 | 0.5569 |
| MP@200 | H3K4me3 | **0.7051** | 0.6369 | 0.6377 |
| | H3K27me3 | **0.6191** | 0.6112 | 0.5254 |
| | H3K9me3 | 0.4796 | 0.4664 | **0.4954** |
| | H3K36me3 | **0.6313** | 0.5549 | 0.5415 |

Table 5: Performance of BioSASANet variants on Plant datasets.

| Eval metrics | Datasets | BioSASA | BioSASA w Evo2-7b | BioSASA w HyenaDNA |
|---|---|---|---|---|
| MSE | Plant Leaf | **2.0153** | 2.7044 | 2.0562 |
| | Plant Proto | **1.3017** | 1.5346 | 1.3231 |
| MP@5 | Plant Leaf | 0.1385 | **0.1762** | 0.1080 |
| | Plant Proto | **0.1489** | 0.1452 | 0.1249 |
| MP@10 | Plant Leaf | 0.1450 | **0.1747** | 0.1429 |
| | Plant Proto | **0.1487** | 0.1447 | 0.1394 |
| MP@15 | Plant Leaf | 0.1413 | **0.1723** | 0.1463 |
| | Plant Proto | **0.1463** | 0.1434 | 0.1406 |

mative and stable attributions. These trends suggest that richer intermediate representations in both NMCSM and PSVM improve the model's interpretability quality.

## A.8 CASE STUDY ON PLANT DATASETS

We conduct a case study to evaluate whether the attributions generated by BioSASANet can correctly highlight known regulatory elements within plant promoter sequences. For each dataset, we select three representative promoter sequences and visualize the base-level attribution scores along the full 170-bp input window. In each plot, the height of each nucleotide reflects its contribution to the predicted promoter strength—taller letters indicate higher attribution. At the top of each panel, we display canonical position-weight matrices (PWMs) for major core promoter elements used as references. The black rectangles highlight the experimentally validated locations of core promoter

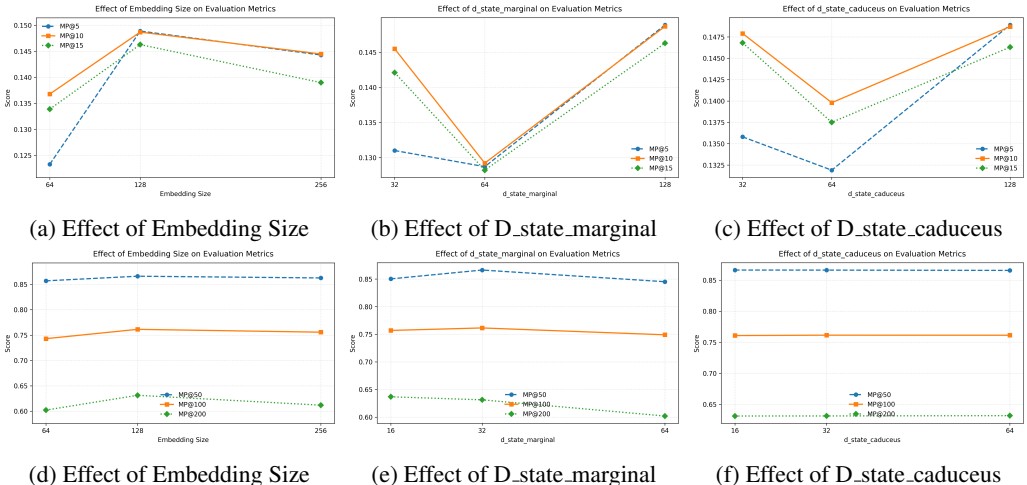

(a) Effect of Embedding Size     (b) Effect of D_state_marginal     (c) Effect of D_state_caduceus

(d) Effect of Embedding Size     (e) Effect of D_state_marginal     (f) Effect of D_state_caduceus

Figure 4: Comparison of parameter experiments on two datasets for three hyperparameters.

elements within each sequence (ground truth). As shown in Figure 5 , BioSASANet produces concentrated high-attribution signals that align precisely with known core promoter elements such as the TATA box and Y-patch, demonstrating that the model not only predicts promoter strength accurately, but also assigns biologically faithful importance to the correct regulatory loci.

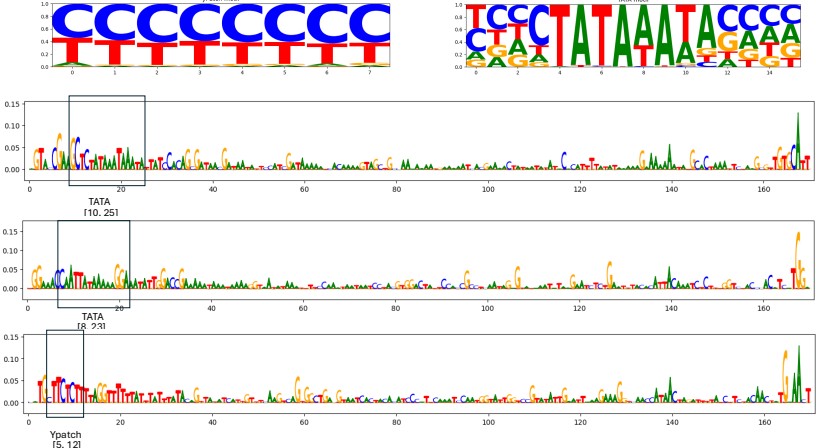

Figure 5: Motif discovery from Shapley values (Leaf).

