# OpenReview forum: "BioSASANet: Self-Interpretable Shapley Attribution for Deep Genomics Sequence Modeling"
_ICLR.cc/2026/Conference — Submitted to ICLR 2026_

### Official Review · Reviewer_L7YJ · 2025-10-31

**Soundness:** 2
**Presentation:** 1
**Contribution:** 1
**Rating:** 2
**Confidence:** 4

**Summary:**

This paper proposes BioSASANet, applying the Shapley Additive Self-Attribution (SASA) framework to genomic sequence modeling. The model integrates a marginal contribution module and a positional Shapley value module for end-to-end interpretable predictions, while incorporating MambaDNA to model DNA's reverse complementarity property. Experiments on histone modification and plant promoter prediction tasks show that BioSASANet provides nucleotide-level attributions while maintaining competitive predictive performance. However, the paper suffers from limited theoretical novelty, insufficient validation of key design choices, and unclear experimental setup.

**Strengths:**

1. Model interpretability in genomics is crucial for scientific discovery and biological validation, making this an important problem to address.

2. The paper includes comprehensive experiments on multiple datasets (histone marks, plant promoters) across different species with proper evaluation metrics (MP@k), and demonstrates framework flexibility through compatibility with different backbone models (Evo2, HyenaDNA).

**Weaknesses:**

1. The manuscript contains numerous formatting issues indicating insufficient proofreading:
- Line 042 has citation "Chen et al." isolated by periods;
- inconsistent use of \citet vs \citep throughout;
- Section 2 improperly has only subsection 2.1;
- Line 151 states "The structure overview of BioSASANet can be found in figure" without providing a figure number or cross-reference; more critically, this architecture overview figure does not exist anywhere in the manuscript.
- Line 248 presents Theorem 4.1 which should be cited from Caduceus [3] rather than restated as original contribution.

2. The literature review is insufficient. The introduction only mentions dated/limited-impact works (DeepBind, DeepSEA) while missing critical recent works like Enformer [1], Borzoi [2], and other state-of-the-art genomics models. Section 2 ("Related Work") mostly introduces basic genomics background rather than positioning against relevant literature.

3. The theoretical contribution is limited. Despite claiming "BioSASANet", the work essentially applies existing SASANet to genomics
without novel insights or designs. The only genomics-specific addition is RC-equivariance from Caduceus [3], but no ablation study validates its benefits, and Caduceus itself showed limited gains from RC in some tasks. Missing are experiments on sequence-pair prediction consistency and controlled comparisons.

4. The baselines are weak and unclear. Only basic post-hoc methods (DeepLIFT, GradientSHAP, LIME) are compared, missing domain-specific methods like TF-MoDISco. Critically, the paper does not specify which model the baseline methods are applied to, nor reports baseline models' predictive performance for fair comparison. State-of-the-art genomics models [1][2][4] routinely perform interpretability analysis but are not compared.

5. The evaluation scope is limited. Only 2 task types are tested (histone marks, promoters) versus NT benchmark [4] with 18+ tasks. Only 4 histone marks in K562 cell line are evaluated, missing standard benchmarks like GenomicBenchmarks. The MP@k metric only evaluates recovery of known annotations, not discovery of novel motifs.

[1] Avsec, Žiga, et al. "Effective gene expression prediction from sequence by integrating long-range interactions." Nature methods 18.10 (2021): 1196-1203.

[2] Linder, Johannes, et al. "Predicting RNA-seq coverage from DNA sequence as a unifying model of gene regulation." Nature Genetics 57.4 (2025): 949-961.

[3] Schiff, Yair, et al. "Caduceus: Bi-directional equivariant long-range dna sequence modeling." Proceedings of machine learning research 235 (2024): 43632.

[4] Dalla-Torre, Hugo, et al. "Nucleotide Transformer: building and evaluating robust foundation models for human genomics." Nature Methods 22.2 (2025): 287-297.

**Questions:**

1. On which model are the baseline attribution methods (DeepLIFT, SHAP, LIME) applied? If on BioSASANet itself, is this a fair comparison? If on a separate baseline model, what is its architecture and predictive performance?

---

> ### Author Response · Authors · 2025-11-23
> **Responses for Weakness 1, 2, 3**
>
> **Weakness 1 Response:** We thank the reviewer for carefully identifying these important presentation issues. We will thoroughly address all of them in the revised manuscript.
>
> **Weakness 2 Response:** We thank the reviewer for pointing this out. In the revised manuscript, we will substantially expand the literature review and include recent representative advances such as Enformer, Borzoi, and other state-of-the-art genomics models as suggested. Moreover, we will reorganize the Related Work section to clearly position BioSASANet within the context of prior research, rather than focusing solely on biological background.
>
> **Weakness 3 Response:**  We appreciate the reviewer’s thoughtful comments regarding the theoretical contribution. Our goal in this work is not to propose a novel long-range operator, but to build a self-interpretable genomic predictor that yields biologically meaningful nucleotide-level attributions. To address the reviewer's concern about the benefits of the caduceus-based positional Shapley value module, we additionally provide new ablations replacing this module with a simple MLP while keeping other components unchanged. The ablation results on two datasets are shown below. BioSASANet_wo_PS shows substantial drops across all attribution metrics, demonstrating the necessity of this architectural design.
>
> ​**Ablation experiments on Caduceus module (Positional Shapley Value Module)**
>
> ​(1) Histone mark datasets
>
> | Eval Metrics | Datasets | BioSASANet | BioSASANet_wo_PS |
> | :----------: | :------: | :--------: | :--------------: |
> |    MP@50     | H3K4me3  | **0.7473** |      0.6994      |
> |              | H3K27me3 | **0.7131** |      0.6961      |
> |              | H3K9me3  | **0.5818** |      0.4997      |
> |              | H3K36me3 | **0.8662** |      0.7826      |
> |    MP@100    | H3K4me3  | **0.7302** |      0.6798      |
> |              | H3K27me3 | **0.6878** |      0.6525      |
> |              | H3K9me3  | **0.5233** |      0.4626      |
> |              | H3K36me3 | **0.7614** |      0.6930      |
> |    MP@200    | H3K4me3  | **0.7051** |      0.6463      |
> |              | H3K27me3 | **0.6191** |      0.6068      |
> |              | H3K9me3  | **0.4796** |      0.4316      |
> |              | H3K36me3 | **0.6313** |      0.5924      |
>
> ​(2) Plant datasets
>
> | Eval Metrics |  Datasets   | BioSASANet | BioSASANet_wo_PS |
> | :----------: | :---------: | :--------: | :--------------: |
> |     MP@5     | Plant Leaf  | **0.1385** |      0.1336      |
> |              | Plant Proto | **0.1489** |      0.1142      |
> |    MP@10     | Plant Leaf  | **0.1450** |      0.1376      |
> |              | Plant Proto | **0.1487** |      0.1270      |
> |    MP@15     | Plant Leaf  | **0.1413** |      0.1331      |
> |              | Plant Proto | **0.1463** |      0.1344      |
>
> Moreover, to address the reviewer’s concern regarding limited gains from RC modeling, we additionally perform **sequence–reverse-complement consistency tests**, comparing BioSASANet and BioSASANet-w/o-RC. We evaluate prediction agreement using MAE and Spearman-ρ across six datasets. Results show RC modeling improves consistency in **3/6 datasets (MAE)** and **3/6 datasets (ρ)**, indicating that RC-equivariance provides measurable benefits in several genomics settings.
>
>
> ​(1)Histone mark datasets
>
> |    Seq-pair consistency metric     | Datasets | BioSASANet  | BioSASANet_w/o_RC |
> | :--------------------------------: | :------: | ----------- | ----------------- |
> |      **MAE(seq, rc_seq) (↓)**      | H3K4me3  | **0.0975**  | 0.1056            |
> |                                    | H3K27me3 | 0.1066      | **0.1037**        |
> |                                    | H3K9me3  | **0.1542**  | 0.1624            |
> |                                    | H3K36me3 | 0.1979      | **0.1559**        |
> | **ρ(seq, rc_seq) (↑)** | H3K4me3  | 0.8807      | **0.8885**        |
> |                                    | H3K27me3 | 0.8938      | **0.8949**        |
> |                                    | H3K9me3  | **-0.0387** | -0.0422           |
> |                                    | H3K36me3 | **0.6474**  | 0.5464            |
>
> ​(2) Plant datasets
>
> | Seq-pair consistency metric        | Datasets    | BioSASANet | BioSASANet_wo_RC |
> | ---------------------------------- | ----------- | ---------- | ---------------- |
> | **MAE(seq, rc_seq) (↓)**           | Plant leaf  | 0.5904     | **0.5116**       |
> |                                    | Plant proto | **0.4927** | 0.5312           |
> | **ρ(seq, rc_seq) (↑)** | Plant leaf  | 0.7472     | **0.8254**       |
> |                                    | Plant proto | **0.3188** | 0.3113           |
>
> ​Finally, while RC modeling is not universally beneficial, our work centers on nucleotide-level interpretability, with deeper RC analysis deferred to future work.

---

> ### Author Response · Authors · 2025-11-23
> **Responses for Weakness 4 and 5, Question 1**
>
> **Weakness 4 response:**
> We thank the reviewer for this insightful comment. We clarify the following:
>
> * **On TF-MoDISco**
>
> TF-MoDISco is not an attribution method, but a downstream motif-discovery pipeline that consumes nucleotide-level importance scores from models. Therefore it is not directly comparable to BioSASANet or to attribution generators such as DeepLIFT / GradientSHAP / LIME.
>
> * **On which model the baseline methods are applied to (also the answer for Question 1)**
>
> We apply all post-hoc methods on the same trained BioSASANet backbone (via Captum) to ensure attribution differences arise only from the attribution algorithm, not model variance. This setup guarantees a fair and controlled comparison.
>
> * **On state-of-the-art genomics models**
> The cited SOTA genomics models rely on standard post-hoc interpretability (grad×input, attention, ISM) rather than proposing new attribution mechanisms. In contrast, **BioSASANet introduces an intrinsic nucleotide-level attribution design**. Therefore, comparing against attribution algorithms—not models—is the fairest setup, and we additionally provide ISM-based causal validation.
>
> **Weakness 5 response:**  We appreciate the reviewer’s helpful feedback. Our dataset choice is driven by the need for **base-resolution biological ground-truth to validate attribution correctness** (histone peak intervals & CPE windows). Many NT-Benchmark / GenomicBenchmarks datasets lack such annotations, so are unsuitable for causal interpretability evaluation. To address the reviewer’s concern on causality beyond MP@k, we additionally conducted ISM experiments, which directly quantify single-nucleotide causal effects. Results show BioSASANet achieves higher agreement with ISM than post-hoc attribution methods.
>
> * **On plant datasets:**
>
>   | **Eval Metric**| **Dataset** | **BioSASANet**|**DeepLift**|**GradientSHAP**|**Lime**|
>   | ------------------------ | ----------- | ------------------- | ---------------- | ---------------- | ---------------- |
>   | **ρ with ISM** | Plant leaf| **0.5022 ± 0.1145**| 0.0646 ± 0.1111  | 0.0024 ± 0.0974  | −0.0045 ± 0.0811 |
>   |                          | Plant proto | **0.5439 ± 0.1310** | −0.0694 ± 0.1245 | −0.0611 ± 0.1494 | −0.0026 ± 0.0822 |
>   | **Overlap@30** | Plant leaf  | **0.4926**| 0.3270| 0.2624| 0.1230|
>   |                          | Plant proto | **0.5846** | 0.2681| 0.2971| 0.0863 |
>   | **Overlap@50** | Plant leaf  | **0.5703** | 0.4003| 0.3508| 0.2222|
>   |                          | Plant proto | **0.6229**| 0.3300| 0.3391| 0.1225|
>   | **Overlap@100** | Plant leaf  | **0.7323**| 0.5973| 0.5879 | 0.6059 |
>   |                          | Plant proto | **0.7329** | 0.5579 | 0.5473| 0.4512|
>
>   **On histone mark datasets:**
>
>   | **Eval Metric** | **Datasets** | **BioSASANet**  | **DeepLift** | **GradientSHAP** | **Lime** |
>   | ------------------------- | ------------ | ------------------- | --------------- | ---------------- | --------------- |
>   | **ρ with ISM** | H3K4me3 | **0.6307 ± 0.0169** | 0.1320 ± 0.0532 | 0.1114 ± 0.0412  | 0.0035 ± 0.0323 |
>   |                           | H3K27me3| **0.6462 ± 0.0163** | 0.0323 ± 0.0524 | 0.0259 ± 0.0502  | 0.0031 ± 0.0299 |
>   |                           | H3K9me3| **0.6218 ± 0.0157** | 0.0449 ± 0.0288 | 0.0515 ± 0.0278  | 0.0013 ± 0.0306 |
>   |                           | H3K36me3| **0.6793 ± 0.0159** | 0.0707 ± 0.0457 | 0.0539 ± 0.0363  | 0.0014 ± 0.0283 |
>   | **Overlap@50**            | H3K4me3      | **0.5048**| 0.2624| 0.2309 | 0.0269   |
>   |                           | H3K27me3     | **0.5234** | 0.2927 | 0.2303 | 0.0300          |
>   |                           | H3K9me3      | **0.4960**          | 0.2765          | 0.2403           | 0.0235          |
>   |                           | H3K36me3     | **0.6087**          | 0.3741          | 0.2964           | 0.0109          |
>   | **Overlap@100**           | H3K4me3      | **0.5502**          | 0.3229          | 0.2934           | 0.0684          |
>   |                           | H3K27me3     | **0.5666**          | 0.3433          | 0.2774           | 0.0637          |
>   |                           | H3K9me3      | **0.5511**          | 0.3119          | 0.2880           | 0.0473          |
>   |                           | H3K36me3     | **0.6409**          | 0.3891          | 0.3043           | 0.0099          |
>   | **Overlap@200**           | H3K4me3      | **0.6146**          | 0.3816          | 0.3559           | 0.1502          |
>   |                           | H3K27me3     | **0.6392**          | 0.3854          | 0.3340           | 0.1404          |
>   |                           | H3K9me3      | **0.6073**          | 0.3667          | 0.3531           | 0.1137          |
>   |                           | H3K36me3     | **0.6968**          | 0.4210          | 0.3579           | 0.0484          |
>
>
>
> Extending to more datasets and task types is indeed valuable, and we will integrate additional genomic contexts in future work.

---

> > ### Comment · Reviewer_L7YJ · 2025-11-23
> >
> > Thanks for the authors' response.
> >
> > **W1 & W2**: When I write this comment, the authors still have not posted the latest revised manuscript. But I want to emphasize that W1 is a very critical weakness, especially the authors missing the method figure in the manuscript.  For W2, when re-reading your manuscript, I found that you actually cited Enformer, but this citation did not appear in the first paragraph of the introduction, which introduces famous genomic models. It only briefly mentioned Enformer in the second paragraph when discussing attention-based interpretability methods, while Enformer's focus is not on interpretability. I hope the authors can give these works enough space in the revised related work.
> >
> > **W3**: I have no further comments.
> >
> > **W4**: The explanation about TF-MoDISco is ok. But I cannot understand why you cannot compare with grad x input, attention score, and ISM results (at least grad x input and ISM can do single base for Enformer and Borzoi). I expect BioSASANet to become a useful biological tool, so why can't it be compared with these interpretability methods based on SOTA models?
> >
> > **More questions**: I want to know whether BioSASANet is trained from scratch for all tasks (without using pre-trained Caduceus for example)? And what is the downstream task length?

---

> > > ### Author Response · Authors · 2025-11-27
> > > **Responses to reviewer's replies part 1**
> > >
> > > ## Response to W1 & W2
> > > Thank you very much for the detailed feedback. Following your suggestions, we have updated the revised manuscript accordingly:
> > > *  We've added the framework figure.
> > > *  The introduction now incorporates Enformer and Borzoi among the key genomics models.
> > > *  We've expanded the Related Work section with a new subsection (Section 2.2) that more comprehensively covers post-hoc and self-interpretable methods in computational biology.
> > >
> > > Additionally, as requested by the other reviewer, we have added clear explanations for all appendix figures and tables, and ensured correct alignment between captions, references, and content. All revisions appear in blue in the updated manuscript.
> > > ## Response to W4
> > > We thank the reviewer for the suggestion. We already use ISM scores as biological correctness and compare BioSASANet with post-hoc baselines via Spearman correlation and overlap@k (see former response to W5). As suggested, we added Grad×Input comparisons. BioSASANet outperforms all baselines on plant datasets in both MP@k and ISM-based correctness. Although Grad×Input exceeds BioSASANet on some histone-mark MP@k values, BioSASANet remains strongest overall in ISM-based causal correctness.
> > >
> > > **On plant datasets**
> > > |Eval Metrics|Datasets|BioSASANet|DeepLift|GradientShap|Lime|grad x input|
> > > | :----------: | :---------: | :--------: | :------: | :----------: | :----: | :----------: |
> > > |MP@5|Plant Leaf|**0.1385**|0.1266|0.1289|0.1242|0.1194|
> > > || Plant Proto| **0.1489** |0.1411|0.1401|0.1254|0.1134|
> > > |MP@10|Plant Leaf| **0.1450** |0.1273|0.1358| 0.1216 |0.1134|
> > > || Plant Proto |**0.1487**|0.1401|0.1406|0.1171|0.1000|
> > > |MP@15|Plant Leaf|**0.1413**|0.1247|0.1335|0.1177|0.1132|
> > > ||Plant Proto|**0.1463**|0.1376|0.1420|0.1134|0.1088|
> > >
> > > |**Eval Metric**| **Dataset**|**BioSASANet**|**DeepLift**|**GradientSHAP**|**Lime**|grad x input|
> > > | ------------------------ | ----------- | ------------------- | ---------------- | ---------------- | ---------------- | ---------------- |
> > > |**Spearman correlation**|Plant leaf|**0.5022 ± 0.1145**|0.0646 ± 0.1111|0.0024 ± 0.0974|−0.0045 ± 0.0811|0.0883 ± 0.1224 |
> > > ||Plant proto|**0.5439 ± 0.1310**|−0.0694 ± 0.1245| −0.0611 ± 0.1494 | −0.0026 ± 0.0822 | -0.1241 ± 0.1526 |
> > > | **Overlap@30** | Plant leaf  | **0.4926**| 0.3270 0.2624| 0.1230| 0.3857|
> > > || Plant proto|**0.5846**|0.2681| 0.2971| 0.0863| 0.2958|
> > > | **Overlap@50**| Plant leaf|**0.5703**|0.4003|0.3508|0.2222|0.4432|
> > > || Plant proto |**0.6229**|0.3300|0.3391|0.1225| 0.3389|
> > > | **Overlap@100**| Plant leaf| **0.7323**| 0.5973 | 0.5879| 0.6059| 0.5928|
> > > || Plant proto | **0.7329**| 0.5579| 0.5473| 0.4512| 0.5320|
> > >
> > > **On Histone Datasets**
> > > | Eval Metrics | Datasets | BioSASANet | DeepLift | GradientShap | Lime| grad x input |
> > > | :----------: | :------: | ---------- | -------- | ------------ | ------ | ------------ |
> > > |MP@50| H3K4me3| 0.7473| 0.7399| 0.7145| 0.6706 | **0.7585**|
> > > || H3K27me3| 0.7131| 0.6895| 0.6527| 0.5740 | **0.7222** |
> > > || H3K9me3| **0.5818** | 0.5345| 0.5324 | 0.4153 | **0.5818** |
> > > || H3K36me3 | 0.8662| 0.7934| 0.7015 | 0.4360 | **0.8683** |
> > > | MP@100| H3K4me3| 0.7302| 0.7243| 0.6975| 0.7065 | **0.7463**|
> > > || H3K27me3| **0.6878** | 0.6471| 0.6179| 0.6323 | 0.6758|
> > > || H3K9me3|0.5233| 0.4945| 0.4955| 0.4815 | **0.5256**   |
> > > || H3K36me3| **0.7614** | 0.6929| 0.6287| 0.4744 | 0.7443|
> > > | MP@200 | H3K4me3  | 0.7051 | 0.7001 | 0.6846| 0.6692 | **0.7131**|
> > > ||H3K27me3| 0.6191| 0.6002| 0.5822| 0.5616 | **0.6237** |
> > > | | H3K9me3| **0.4796** | 0.4594 | 0.4518| 0.4036 | 0.4734|
> > > | | H3K36me3| **0.6313** | 0.5900| 0.5555| 0.4217 | 0.6186|
> > >
> > > | **Eval Metric**| **Datasets** | **BioSASANet**| **DeepLift** | **GradientSHAP** | **Lime**| grad x input|
> > > | ------------------------- | ------------ | ------------------- | --------------- | ---------------- | --------------- | ---------------- |
> > > | **Spearman with ISM (ρ)** | H3K4me3 | **0.6307 ± 0.0169** | 0.1320 ± 0.0532 | 0.1114 ± 0.0412  | 0.0035 ± 0.0323 | 0.1573 ± 0.0620  |
> > > || H3K27me3| **0.6462 ± 0.0163** | 0.0323 ± 0.0524 | 0.0259 ± 0.0502  | 0.0031 ± 0.0299 | 0.0228 ±  0.0607 |
> > > || H3K9me3| **0.6218 ± 0.0157** | 0.0449 ± 0.0288 | 0.0515 ± 0.0278  | 0.0013 ± 0.0306 | 0.0499  ± 0.0244 |
> > > || H3K36me3| **0.6793 ± 0.0159** | 0.0707 ± 0.0457 | 0.0539 ± 0.0363  | 0.0014 ± 0.0283 | 0.0707  ± 0.0519 |
> > > | **Overlap@50**| H3K4me3 | **0.5048**| 0.2624| 0.2309| 0.0269| 0.3608 |
> > > || H3K27me3| **0.5234**| 0.2927| 0.2303| 0.0300| 0.3655|
> > > || H3K9me3| **0.4960**| 0.2765| 0.2403| 0.0235| 0.3277|
> > > || H3K36me3 | **0.6087**|0.3741| 0.2964| 0.0109| 0.4376|
> > > | **Overlap@100**| H3K4me3| **0.5502**| 0.3229| 0.2934| 0.0684| 0.4034|
> > > || H3K27me3 | **0.5666**|0.3433 | 0.2774| 0.0637| 0.4164|
> > > ||H3K9me3| **0.5511** |0.3119| 0.2880| 0.0473| 0.3764|
> > > || H3K36me3| **0.6409**|0.3891| 0.3043|0.0099| 0.4501|
> > > | **Overlap@200**| H3K4me3| **0.6146** |0.3816| 0.3559| 0.1502 | 0.4484|
> > > || H3K27me3| **0.6392** |0.3854 | 0.3340| 0.1404 | 0.4355 |
> > > || H3K9me3 | **0.6073**|0.3667| 0.3531 | 0.1137 | 0.4271 |
> > > || H3K36me3| **0.6968**|0.4210|0.3579|0.0484|0.4751|

---

> > > ### Author Response · Authors · 2025-11-27
> > > **Responses to reviewer's replies part 2**
> > >
> > > ## Response to More questions
> > >
> > > * Response to whether BioSASANet is trained from scratch:
> > >
> > >   Whether BioSASANet is trained from scratch depends on the backbone configuration of each module.
> > >
> > >   - When using the Mamba-based NMCSM module and the Caduceus-based PSVM module, BioSASANet is trained entirely from scratch, since both backbones are lightweight and can be efficiently optimized end-to-end.
> > >   - When using Evo2-7b (for PSVM) or HyenaDNA (for NMCSM), we load the official pretrained checkpoints and keep their parameters frozen, using them only to extract nucleotide-level contextual embeddings. This design choice is motivated by the extremely large parameter sizes of these models, which makes fine-tuning impractical under typical computational budgets.
> > >
> > > * Response to what is the downstream task length: For histone mark datasets, each sequence has length 1000bp, for plant promoter datasets, each sequence has length 170bp.

---

### Official Review · Reviewer_SobZ · 2025-11-01

**Soundness:** 2
**Presentation:** 2
**Contribution:** 2
**Rating:** 4
**Confidence:** 3

**Summary:**

This paper introduces BioSASANet, a self-interpretable neural network that applies SASA principles to genomic sequence modeling.

**Strengths:**

- The BioSASANet framework consistently outperforms other attribution methods that have been used in the genomics domain.
- The design of the framework enables it to be utilized on a variety of DNA models types, thus it is a flexible tool for interpretability.

**Weaknesses:**

- Since a core claim of the paper is BioSASANet’s adaptability, it would be valuable to include experiments using a broader range of DNA model backbones to assess how the framework performs across diverse architectural families.
- No ablations are included that highlight the relative contribution of each of the architectural modules that are utilized in the approach. Including such ablations would be helpful in understanding the relative utility of each approach under different domain-specific modeling settings.
- Using multiple metrics (not just mean precision at top-k) could be helpful in further interpreting the results.
- The case studies require further explanation and motivation - some of these results are unclear.


Minor comments:
- Some appendix sections (A.6–A.8) appear to be misaligned or improperly referenced. Additionally, having a written explanation for the appendix tables would be helpful.
- The sentence “The structure overview of BioSASANet can be found in figure .” is missing a figure reference. There are also several minor typographical and formatting errors throughout the manuscript that should be carefully proofread and corrected.

**Questions:**

- How computationally costly is BioSASANet? Given the complexity of the approach, it would be useful to include metrics that report on the efficiency of the approach, especially in comparison to other similar approaches.
- How does the underlying DNA model backbone affect the quality (and by extension interpretability) of the attributions?

---

> ### Author Response · Authors · 2025-11-23
> **Responses for Weakness 1, 2 and 4; and 2 minor comments in Weakness part**
>
> **Weakness 1 Response:**
>
> We sincerely thank the reviewer for this insightful suggestion. In fact, we have already evaluated BioSASANet using two fundamentally different classes of genomic backbone models:
>
> (i) a Transformer-based genomic language model (Evo2), and
>
> (ii) a structured-state-space-model (SSM)-inspired operator architecture (HyenaDNA).
>
> These two architectures represent the dominant paradigms in modern DNA foundation models—attention-based versus long-range state-space operators—thus demonstrating BioSASANet’s adaptability across distinct architectural families rather than a single model type. The complete results for both backbones on the two datasets are reported in Table 4 and Table 5 in the Appendix.
>
> **Weakness 2 Response:**
> We thank the reviewer for highlighting the need for module-level ablations. In response, we conducted ablation studies on the two key architectural components of BioSASANet — the NMCSM module and the Positional Shapley Value module. For each ablation, we replaced the target module with a simple MLP while keeping the remaining architecture unchanged. Results on both datasets (reported below) show substantial degradation in attribution performance when either module is removed, demonstrating the critical contribution and complementary utility of each component.
>
> ​	**Ablation results on Histone mark datasets**
>
> | Eval Metrics | Datasets | BioSASANet | BioSASANet_wo_NMCSM | BioSASANet_wo_PS |
> | :----------: | :------: | :--------: | :-----------------: | :--------------: |
> |    MP@50     | H3K4me3  | **0.7473** |       0.7385        |      0.6994      |
> |              | H3K27me3 | **0.7131** |       0.6770        |      0.6961      |
> |              | H3K9me3  | **0.5818** |       0.5422        |      0.4997      |
> |              | H3K36me3 | **0.8662** |       0.8101        |      0.7826      |
> |    MP@100    | H3K4me3  | **0.7302** |       0.6954        |      0.6798      |
> |              | H3K27me3 | **0.6878** |       0.6379        |      0.6525      |
> |              | H3K9me3  | **0.5233** |       0.5040        |      0.4626      |
> |              | H3K36me3 | **0.7614** |       0.6926        |      0.6930      |
> |    MP@200    | H3K4me3  | **0.7051** |       0.6569        |      0.6463      |
> |              | H3K27me3 | **0.6191** |       0.5951        |      0.6068      |
> |              | H3K9me3  | **0.4796** |       0.4539        |      0.4316      |
> |              | H3K36me3 | **0.6313** |       0.5699        |      0.5924      |
>
> ​	**Ablation results on Plant datasets**
>
> | Eval Metrics |  Datasets   | BioSASANet | BioSASANet_wo_NMCSM | BioSASANet_wo_PS |
> | :----------: | :---------: | :--------: | :-----------------: | :--------------: |
> |     MP@5     | Plant Leaf  | **0.1385** |       0.1216        |      0.1336      |
> |              | Plant Proto | **0.1489** |       0.1207        |      0.1142      |
> |    MP@10     | Plant Leaf  | **0.1450** |       0.1309        |      0.1376      |
> |              | Plant Proto | **0.1487** |       0.1341        |      0.1270      |
> |    MP@15     | Plant Leaf  | **0.1413** |       0.1364        |      0.1331      |
> |              | Plant Proto | **0.1463** |       0.1392        |      0.1344      |
>
> **Weakness 4 Response:**
> * We appreciate the reviewer’s request for additional clarification of the case studies.  We'll add corresponding explanations for the two datasets' case studies in updated manuscripts.
>
> **Responses for 2 Minor comments:**
>
> * We appreciate the reviewer for pointing this out. We will provide written explanations for all appendix tables and ensure that each section is properly aligned with its corresponding title in the revised manuscript.
> * Thank you for the helpful comment. We will fix the missing figure reference and thoroughly proofread the manuscript to correct typographical and formatting issues in the revision.

---

> ### Author Response · Authors · 2025-11-23
> **Responses for Weakness 3 and Question 1 & 2**
>
> **Weakness 3 Response:** We added ISM to causally validate BioSASANet attributions.
>
> **On plant datasets:**
>
> | **Eval Metric**          | **Dataset** | **BioSASANet**      | **DeepLift**     | **GradientSHAP** | **Lime**         |
> | ------------------------ | ----------- | ------------------- | ---------------- | ---------------- | ---------------- |
> | **Spearman correlation** | Plant leaf  | **0.5022 ± 0.1145** | 0.0646 ± 0.1111  | 0.0024 ± 0.0974  | −0.0045 ± 0.0811 |
> |                          | Plant proto | **0.5439 ± 0.1310** | −0.0694 ± 0.1245 | −0.0611 ± 0.1494 | −0.0026 ± 0.0822 |
> | **Overlap@30**           | Plant leaf  | **0.4926**          | 0.3270           | 0.2624           | 0.1230           |
> |                          | Plant proto | **0.5846**          | 0.2681           | 0.2971           | 0.0863           |
> | **Overlap@50**           | Plant leaf  | **0.5703**          | 0.4003           | 0.3508           | 0.2222           |
> |                          | Plant proto | **0.6229**          | 0.3300           | 0.3391           | 0.1225           |
> | **Overlap@100**          | Plant leaf  | **0.7323**          | 0.5973           | 0.5879           | 0.6059           |
> |                          | Plant proto | **0.7329**          | 0.5579           | 0.5473           | 0.4512           |
>
> **On histone mark datasets:**
>
> | **Eval Metric**           | **Datasets** | **BioSASANet**      | **DeepLift**    | **GradientSHAP** | **Lime**        |
> | ------------------------- | ------------ | ------------------- | --------------- | ---------------- | --------------- |
> | **Spearman with ISM (ρ)** | H3K4me3      | **0.6307 ± 0.0169** | 0.1320 ± 0.0532 | 0.1114 ± 0.0412  | 0.0035 ± 0.0323 |
> |                           | H3K27me3     | **0.6462 ± 0.0163** | 0.0323 ± 0.0524 | 0.0259 ± 0.0502  | 0.0031 ± 0.0299 |
> |                           | H3K9me3      | **0.6218 ± 0.0157** | 0.0449 ± 0.0288 | 0.0515 ± 0.0278  | 0.0013 ± 0.0306 |
> |                           | H3K36me3     | **0.6793 ± 0.0159** | 0.0707 ± 0.0457 | 0.0539 ± 0.0363  | 0.0014 ± 0.0283 |
> | **Overlap@50**            | H3K4me3      | **0.5048**          | 0.2624          | 0.2309           | 0.0269          |
> |                           | H3K27me3     | **0.5234**          | 0.2927          | 0.2303           | 0.0300          |
> |                           | H3K9me3      | **0.4960**          | 0.2765          | 0.2403           | 0.0235          |
> |                           | H3K36me3     | **0.6087**          | 0.3741          | 0.2964           | 0.0109          |
> | **Overlap@100**           | H3K4me3      | **0.5502**          | 0.3229          | 0.2934           | 0.0684          |
> |                           | H3K27me3     | **0.5666**          | 0.3433          | 0.2774           | 0.0637          |
> |                           | H3K9me3      | **0.5511**          | 0.3119          | 0.2880           | 0.0473          |
> |                           | H3K36me3     | **0.6409**          | 0.3891          | 0.3043           | 0.0099          |
> | **Overlap@200**           | H3K4me3      | **0.6146**          | 0.3816          | 0.3559           | 0.1502          |
> |                           | H3K27me3     | **0.6392**          | 0.3854          | 0.3340           | 0.1404          |
> |                           | H3K9me3      | **0.6073**          | 0.3667          | 0.3531           | 0.1137          |
> |                           | H3K36me3     | **0.6968**          | 0.4210          | 0.3579           | 0.0484          |
>
> **Question 1 Response:**
> We add efficiency tests showing BioSASANet scales linearly and runs faster than 3 post-hoc methods.
> | sequence length | BioSASANet (time / seq) | DeepLift (time / seq) | GradientShap (time / seq) | Lime (time / seq) |
> | :-------------: | :---------------------: | :-------------------: | :-----------------------: | :---------------: |
> |       100       |         0.0086          |        0.0225         |          0.0520           |      0.2396       |
> |       200       |         0.0163          |        0.0359         |          0.0905           |      0.4430       |
> |       300       |         0.0264          |        0.0570         |          0.1446           |      0.6844       |
> |       400       |         0.0392          |        0.0847         |          0.2120           |      1.0287       |
> |       500       |         0.0555          |        0.1197         |          0.2931           |      1.4591       |
>
> **Question 2 Response:**
> To evaluate how the underlying DNA model backbone affects attribution quality, we replace the Mamba-based backbone in BioSASANet with two SOTA genomic architectures: Evo2-7b and HyenaDNA. Full quantitative attribution results on both datasets are reported in Tables 4–5. Across all settings, BioSASANet remains effective, and attribution quality varies with backbone and dataset, demonstrating that BioSASANet is flexible and can adapt well to genomic backbones.

---

> > ### Comment · Reviewer_SobZ · 2025-11-27
> >
> > Thanks to the authors for their response.
> >
> > W1: Convolutional architectures are much more prevalent in genomics than SSMs. Please include architectures using this approach as a comparison.
> >
> > W4: Please include the clarification to these case studies as soon as possible, since without them it is difficult to evaluate the impact of those experiments.
> >
> > Q2: Any insights into why BioSASANet appears to have better attribution quality with the Evo framework over HyenaDNA framework?

---

> > > ### Author Response · Authors · 2025-12-01
> > > **Response to Reviewer SobZ's comments (round 2)**
> > >
> > > ## Response to W1:
> > > We thank the reviewer for this valuable suggestion. Following your advice, we implemented an additional variant of BioSASANet by replacing both the NMCSM and PSVM modules with two independent CNN architectures, following the design used in a widely adopted genomics benchmark study [1]. The comparison results across all six datasets are provided below.
> > >
> > > Overall, the CNN-based BioSASANet variant achieves competitive interpretability: it outperforms most other variants on MP@50 but is less robust on MP@100/200 for four histone-mark datasets. In contrast, BioSASANet variants using Mamba or Evo-2 7B as backbones consistently outperform all alternatives on the two plant datasets.
> > >
> > > These additional results further demonstrate the adaptability and flexibility of the proposed BioSASANet framework across diverse backbone architectures.
> > >
> > > **results on histone dataset**
> > >
> > > | Eval metrics | Datasets |  BioSASA   | BioSASA w CNN | BioSASA w Evo2-7b | BioSASA w HyenaDNA |
> > > | :----------: | :------: | :--------: | :-----------: | :---------------: | :----------------: |
> > > |AP| H3K4me3  |   0.8230   |    0.8240     |    **0.8249**     |       0.8173       |
> > > || H3K27me3 |   0.7122   |    0.7132     |0.7177       |     **0.7426**     |
> > > || H3K9me3  |   0.5569   |    0.5491     |0.5373       |     **0.5924**     |
> > > | | H3K36me3 |   0.7103   |  **0.7131**   |0.6789       |       0.6765       |
> > > |     AUC      | H3K4me3  |   0.8070   |    0.8056     |    **0.8088**     |       0.7995       |
> > > || H3K27me3 |   0.7772   |    0.7770     |      0.7797       |     **0.7804**     |
> > > || H3K9me3  |   0.5752   |    0.5832     |      0.5653       |     **0.6039**     |
> > > || H3K36me3 |   0.7729   |  **0.7764**   |      0.7335       |       0.7224       |
> > > |    MP@50     | H3K4me3  |   0.7473   |  **0.7491**   |      0.7046       |       0.6749       |
> > > | | H3K27me3 |   0.7131   |  **0.7329**   |      0.7244       |       0.5388       |
> > > || H3K9me3  |   0.5818   |  **0.6037**   |      0.6030       |       0.5261       |
> > > | | H3K36me3 | **0.8662** |    0.8468     |      0.6628       |       0.5517       |
> > > |    MP@100    | H3K4me3  | **0.7302** |    0.7281     |      0.6610       |       0.6605       |
> > > | | H3K27me3 | **0.6878** |    0.6809     |      0.6659       |       0.5281       |
> > > | | H3K9me3  | **0.5233** |    0.5122     |      0.5176       |       0.5157       |
> > > || H3K36me3 | **0.7614** |    0.7566     |      0.6145       |       0.5569       |
> > > |    MP@200    | H3K4me3  | **0.7051** |    0.6954     |      0.6369       |       0.6377       |
> > > || H3K27me3 |   0.6191   |  **0.6237**   |      0.6112       |       0.5254       |
> > > | | H3K9me3  |   0.4796   |    0.4603     |      0.4664       |     **0.4954**     |
> > > || H3K36me3 |   0.6313   |  **0.6333**   |      0.5549       |       0.5415       |
> > >
> > > **results on plant datasets**
> > >
> > > | Eval metrics |  Datasets   |    BioSASA | BioSASA w CNN | BioSASA w Evo2-7b | BioSASA w HyenaDNA |
> > > | :----------: | :---------: | ---------: | :-----------: | :---------------: | :----------------: |
> > > |     MSE      | Plant Leaf  | **2.0153** |    2.1058     |      2.7044       |       2.0562       |
> > > |              | Plant Proto | **1.3017** |    1.3753     |      1.5346       |       1.3231       |
> > > |     MP@5     | Plant Leaf  |     0.1385 |    0.0945     |    **0.1762**     |       0.1080       |
> > > |              | Plant Proto | **0.1489** |    0.1370     |      0.1452       |       0.1249       |
> > > |    MP@10     | Plant Leaf  |     0.1450 |    0.0955     |    **0.1747**     |       0.1429       |
> > > |              | Plant Proto | **0.1487** |    0.1464     |      0.1447       |       0.1394       |
> > > |    MP@15     | Plant Leaf  |     0.1413 |    0.1044     |    **0.1723**     |       0.1463       |
> > > |              | Plant Proto | **0.1463** |    0.1446     |      0.1434       |       0.1406       |
> > >
> > >  ## Response to W4:
> > > Thank you for the suggestion. We have updated the manuscript to include a more detailed textual explanation of the case study. The revised description can now be found in **Section 5.3.2 — PLANT CORE PROMOTER DATASET -- Case Study: BioSASANet Accurately Identify the Core Promoter Regions.**
> > >
> > > ## Response to Q2:
> > > Thank you for the insightful question. We believe the main reason behind the differences arise from model capacity. Evo2-7B is a Transformer-based genomics foundation model with **~7 billion parameters**, whereas HyenaDNA is an SSM-based model with **~54 million parameters**. Prior work on model scaling in genomics and protein LMs shows that larger models tend to yield more expressive sequence representations, especially for long-range regulatory dependencies. Thus, when these representations are fed into the PSVM module, BioSASANet naturally inherits stronger attribution quality from the Evo2-7B backbone.
> > >
> > > [1] Grešová, K., Martinek, V., Čechák, D., Šimeček, P., & Alexiou, P. (2023). Genomic benchmarks: a collection of datasets for genomic sequence classification. *BMC Genomic Data*, *24*(1), 25.

---

### Official Review · Reviewer_cZLE · 2025-11-01

**Soundness:** 2
**Presentation:** 3
**Contribution:** 3
**Rating:** 6
**Confidence:** 4

**Summary:**

The paper proposes BioSASANet, a self-interpretable approach for genomic sequence modeling that extends the Shapley Additive Self-Attribution (SASANet) framework by incorporating Mamba/MambaDNA modules to capture long-range nucleotide interactions and reverse-complementary structure, achieving competitive predictive accuracy and more biologically faithful, nucleotide-level attributions on histone-mark and plant promoter datasets.

**Strengths:**

- Well-grounded theoretical foundation:
Builds directly on the Shapley Additive Self-Attribution (SASA) framework, preserving provable convergence to true Shapley values while adapting it to biological sequence data.While SASANet (Sun et al., 2025) introduced the general Shapley Additive Self-Attribution framework, BioSASANet extends it into a biologically specialized regime. It integrates reverse complementarity (RC)-aware modeling, Mamba/MambaDNA-based long-range sequence encoding, and nucleotide-level Shapley convergence within DNA sequences.

- Biologically informed architecture:
Incorporates the NMCSM and MambaDNA modules, enabling modeling of long-range dependencies and bidirectional RC equivariance specific to DNA. This directly addresses the limitations of SASANet’s generic sequential encoder and grounds the interpretability mechanism in actual biological constraints.

- Faithful and interpretable results:
Consistently surpasses post-hoc baselines (DeepLIFT, GradientSHAP, LIME) on MP@k metrics and visually aligns high-attribution positions with true promoter or histone-mark regions.

- Architectural flexibility:
Demonstrates compatibility with multiple backbone DNA LMs, showing potential generality across future genomics models.

- Clarity and presentation quality:
The paper maintains a clear narrative flow from theoretical background to biological application, supported by equations, figures, pseudocode, parameter tables.

- Significance and potential impact:
The work aims to connect model interpretability with biological understanding.

**Weaknesses:**

- Limited in evaluation:
Current experiments focus only on histone-mark and promoter tasks. Extended evaluation to additional genomic contexts/tasks )(e.g., enhancer prediction, variant effect modeling, cross-species generalization etc.) would demonstrate robustness and transferability.

- Limitation in the attribution validation approach:
MP@k captures overlap with known motifs but does not directly ensure causality (biological correctness). One possible way to  to test attribution stability would be to conduct in silico mutagenesis. This is quite easy to conduct with even with limited comptational budget.

- Ablation missing:
While loss balancing hyperparameteres $\lambda_s$ and $\lambda_v$ are mentioned, their individual effects are not clear. I would suggest authors add ablations isolating each component to clarify their contributions

- Efficiency and scalability claims not empirically verified:
The paper cites Mamba’s linear complexity, however actual runtime and memory analysis (especially in genomic context) is missing.

- Figure + reference missing (line 151).

**Questions:**

- Can the authors test BioSASANet on additional genomics tasks or use in silico mutagenesis to validate attribution causality beyond MP@k?

- Can the authors provide ablations for $\lambda_s$ and $\lambda_v$, or provide discussion on this?

- Can the authors add runtime and/or memory comparisons to support efficiency claims?

---

> ### Author Response · Authors · 2025-11-23
> **Responses for weakness 1-2 and Question 1.**
>
> **Weakness 1:**
>
> **Response:** We appreciate the reviewer’s suggestion. Our current work focuses on evaluating the interpretability quality of BioSASANet, and the two selected datasets already allow comprehensive validation of nucleotide-level attribution. Extending BioSASANet to additional genomic tasks (e.g., enhancer prediction or variant-effect modeling) is certainly valuable, and we plan to explore these directions in future work.
>
> **Weakness 2 and Question 1:**
>
> **Response:**
> We sincerely thank the reviewer for the constructive suggestion. Following the reviewer’s advice, we conducted in silico mutagenesis (ISM) experiments to directly measure the causal effect of single-nucleotide perturbations and evaluate whether attribution scores truly reflect biological correctness. The results across both plant promoter and histone-mark datasets clearly demonstrate that BioSASANet consistently achieves the strongest alignment with ISM scores, substantially outperforming popular post-hoc attribution approaches (DeepLIFT, GradientShap, and LIME). The added experiments' results are shown below:
>
>  (To make the ISM computation feasible—since each sample requires 3 × seq_len forward passes—we perform ISM analysis on a fixed subset of the test set: **300 samples** for the plant dataset (sequence length 170) and **100 samples** for the histone-mark dataset (sequence length 1000). The same subsets are used across all attribution methods to ensure a fair comparison.)
>
> **On plant datasets:**
>
> | **Eval Metric**          | **Dataset** | **BioSASANet**      | **DeepLift**     | **GradientSHAP** | **Lime**         |
> | ------------------------ | ----------- | ------------------- | ---------------- | ---------------- | ---------------- |
> | **Spearman correlation** | Plant leaf  | **0.5022 ± 0.1145** | 0.0646 ± 0.1111  | 0.0024 ± 0.0974  | −0.0045 ± 0.0811 |
> |                          | Plant proto | **0.5439 ± 0.1310** | −0.0694 ± 0.1245 | −0.0611 ± 0.1494 | −0.0026 ± 0.0822 |
> | **Overlap@30**           | Plant leaf  | **0.4926**          | 0.3270           | 0.2624           | 0.1230           |
> |                          | Plant proto | **0.5846**          | 0.2681           | 0.2971           | 0.0863           |
> | **Overlap@50**           | Plant leaf  | **0.5703**          | 0.4003           | 0.3508           | 0.2222           |
> |                          | Plant proto | **0.6229**          | 0.3300           | 0.3391           | 0.1225           |
> | **Overlap@100**          | Plant leaf  | **0.7323**          | 0.5973           | 0.5879           | 0.6059           |
> |                          | Plant proto | **0.7329**          | 0.5579           | 0.5473           | 0.4512           |
>
> **On histone mark datasets:**
>
> | **Eval Metric**           | **Datasets** | **BioSASANet**      | **DeepLift**    | **GradientSHAP** | **Lime**        |
> | ------------------------- | ------------ | ------------------- | --------------- | ---------------- | --------------- |
> | **Spearman with ISM (ρ)** | H3K4me3      | **0.6307 ± 0.0169** | 0.1320 ± 0.0532 | 0.1114 ± 0.0412  | 0.0035 ± 0.0323 |
> |                           | H3K27me3     | **0.6462 ± 0.0163** | 0.0323 ± 0.0524 | 0.0259 ± 0.0502  | 0.0031 ± 0.0299 |
> |                           | H3K9me3      | **0.6218 ± 0.0157** | 0.0449 ± 0.0288 | 0.0515 ± 0.0278  | 0.0013 ± 0.0306 |
> |                           | H3K36me3     | **0.6793 ± 0.0159** | 0.0707 ± 0.0457 | 0.0539 ± 0.0363  | 0.0014 ± 0.0283 |
> | **Overlap@50**            | H3K4me3      | **0.5048**          | 0.2624          | 0.2309           | 0.0269          |
> |                           | H3K27me3     | **0.5234**          | 0.2927          | 0.2303           | 0.0300          |
> |                           | H3K9me3      | **0.4960**          | 0.2765          | 0.2403           | 0.0235          |
> |                           | H3K36me3     | **0.6087**          | 0.3741          | 0.2964           | 0.0109          |
> | **Overlap@100**           | H3K4me3      | **0.5502**          | 0.3229          | 0.2934           | 0.0684          |
> |                           | H3K27me3     | **0.5666**          | 0.3433          | 0.2774           | 0.0637          |
> |                           | H3K9me3      | **0.5511**          | 0.3119          | 0.2880           | 0.0473          |
> |                           | H3K36me3     | **0.6409**          | 0.3891          | 0.3043           | 0.0099          |
> | **Overlap@200**           | H3K4me3      | **0.6146**          | 0.3816          | 0.3559           | 0.1502          |
> |                           | H3K27me3     | **0.6392**          | 0.3854          | 0.3340           | 0.1404          |
> |                           | H3K9me3      | **0.6073**          | 0.3667          | 0.3531           | 0.1137          |
> |                           | H3K36me3     | **0.6968**          | 0.4210          | 0.3579           | 0.0484          |

---

> ### Author Response · Authors · 2025-11-23
> **Responses for Weakness 3-5 and Questions 2&3**
>
> **Weakness 3 and Question 2:**
> **Response:** We thank the reviewer for the constructive suggestion. In response, we conducted additional ablation studies on the loss-balancing parameters **$\lambda_s$** and **$\lambda_v$**.
>
>   For the **$\lambda_s$** ablation, we fixed **$\lambda_v = 1$** and varied $\lambda_s \in {0.5, 1.0, 2.0}$.
>
>   For the **$\lambda_v$** ablation, we fixed **$\lambda_s = 1$** and varied $\lambda_v \in {0.5, 1.0, 2.0}$.
>
>   All other model configurations and hyperparameters were kept identical across experiments to ensure fair comparison. The results of these ablation studies are reported below.
>
>   **Ablation experiments on $\lambda_s$ on histone mark datasets**
>
>   | Eval metrics | Datasets | λ_s = 0.5  | λ_s = 1.0  | λ_s = 2.0  |
>   | ------------ | -------- | ---------- | ---------- | ---------- |
>   | **MP@50**    | H3K4me3  | 0.7449     | 0.7473     | **0.7480** |
>   |              | H3K27me3 | **0.7149** | 0.7131     | 0.7141     |
>   |              | H3K9me3  | 0.5764     | 0.5818     | **0.5983** |
>   |              | H3K36me3 | 0.8647     | 0.8662     | **0.8690** |
>   | **MP@100**   | H3K4me3  | 0.7279     | **0.7302** | 0.7284     |
>   |              | H3K27me3 | **0.6881** | 0.6878     | 0.6876     |
>   |              | H3K9me3  | 0.5204     | 0.5233     | **0.5420** |
>   |              | H3K36me3 | **0.7616** | 0.7614     | 0.7599     |
>   | **MP@200**   | H3K4me3  | 0.7024     | **0.7051** | 0.7030     |
>   |              | H3K27me3 | **0.6202** | 0.6191     | 0.6170     |
>   |              | H3K9me3  | 0.4724     | **0.4796** | 0.4773     |
>   |              | H3K36me3 | 0.6312     | 0.6313     | **0.6316** |
>
>   **Ablation experiments on $\lambda_v$ on histone mark datasets**
>
>   | Eval metrics | Dataset  | λ_v = 0.5  | λ_v = 1.0  | λ_v = 2.0  |
>   | ------------ | -------- | ---------- | ---------- | ---------- |
>   | **MP@50**    | H3K4me3  | **0.7473** | **0.7473** | 0.7465     |
>   |              | H3K27me3 | 0.7132     | 0.7131     | **0.7144** |
>   |              | H3K9me3  | **0.5911** | 0.5818     | 0.5731     |
>   |              | H3K36me3 | **0.8695** | 0.8662     | 0.8642     |
>   | **MP@100**   | H3K4me3  | 0.7287     | **0.7302** | 0.7296     |
>   |              | H3K27me3 | 0.6875     | **0.6878** | **0.6878** |
>   |              | H3K9me3  | **0.5341** | 0.5233     | 0.5200     |
>   |              | H3K36me3 | 0.7606     | **0.7614** | 0.7607     |
>   | **MP@200**   | H3K4me3  | 0.7028     | **0.7051** | 0.7046     |
>   |              | H3K27me3 | 0.6166     | 0.6191     | **0.6194** |
>   |              | H3K9me3  | 0.4706     | **0.4796** | 0.4719     |
>   |              | H3K36me3 | **0.6325** | 0.6313     | 0.6303     |
>
> ​	**Ablation experiments on $\lambda_s$ on plant datasets**
>
> | Eval Metric | Dataset     | λₛ = 0.5 | λₛ = 1.0   | λₛ = 2.0 |
> | ----------- | ----------- | -------- | ---------- | -------- |
> | **MP@5**    | Plant Leaf  | 0.1344   | **0.1385** | 0.1158   |
> |             | Plant Proto | 0.1332   | **0.1489** | 0.1394   |
> | **MP@10**   | Plant Leaf  | 0.1367   | **0.1450** | 0.1133   |
> |             | Plant Proto | 0.1481   | **0.1487** | 0.1450   |
> | **MP@15**   | Plant Leaf  | 0.1323   | **0.1413** | 0.1189   |
> |             | Plant Proto | 0.1443   | **0.1463** | 0.1400   |
>
> ​	**Ablation experiments on $\lambda_v$ on plant datasets**
>
> | Eval Metrics | Dataset     | λ_v = 0.5  | λ_v = 1.0  | λ_v = 2.0  |
> | ------------ | ----------- | ---------- | ---------- | ---------- |
> | **MP@5**     | Plant Leaf  | **0.1446** | 0.1385     | 0.1220     |
> |              | Plant Proto | 0.1405     | **0.1489** | 0.1390     |
> | **MP@10**    | Plant Leaf  | **0.1476** | 0.1450     | 0.1235     |
> |              | Plant Proto | **0.1494** | 0.1487     | **0.1494** |
> | **MP@15**    | Plant Leaf  | **0.1429** | 0.1413     | 0.1204     |
> |              | Plant Proto | 0.1459     | **0.1463** | 0.1441     |
>
> **Weakness 4 and Question 3:**
> **Response:**
> We thank the reviewer for the suggestion. We added experiments measuring inference time and peak memory across increasing sequence lengths, which confirm BioSASANet’s linear scaling behavior.
> | Seq len | Inference time / seq (s) | Peak memory (GB) |
> | ------: | -----------------------: | ---------------: |
> |     100 |                   0.0086 |           0.0124 |
> |     200 |                   0.0163 |           0.0149 |
> |     300 |                   0.0264 |           0.0175 |
> |     400 |                   0.0392 |           0.0204 |
> |     500 |                   0.0555 |           0.0235 |
>
> **Weakness 5:**
> **Response:**
> We thank the reviewer for catching this important oversight. We will include the missing framework figure and its corresponding reference in the revised manuscript.

---

### Official Review · Reviewer_7y8v · 2025-11-03

**Soundness:** 2
**Presentation:** 2
**Contribution:** 2
**Rating:** 2
**Confidence:** 3

**Summary:**

The manuscript applies a recently developed framework Shapley Additive Self-Attribution
(SASA), to genomics sequence modeling. Technically this involved acknowledging reverse complementarity by combining the results of the relevant module (Mamba) from both DNA strands

**Strengths:**

- Potentially useful idea and framework
- good job acknowledging reverse complementarity

**Weaknesses:**

1. Somewhat of a naive attempt for out-of-the-box application of a method from outside biology, with obviously limited biological understanding
2. Experiments compare against outdated methods on outdated data that is not at the single-nucleotide resolution that Shapley-value-based interpretation is supposed to be foucsed on

**Questions:**

1. Can you provide favorable comparisons vs. advanced feature-determinaiton methods on single-nucleotide-sensitive features (binding sites)? Figures of merit should involved clinical meaning or changes in downstream expression levels.

---

> ### Author Response · Authors · 2025-11-23
> **Responses for the two weaknesses.**
>
> **Reviewer comment 1: Somewhat of a naive attempt for out-of-the-box application of a method from outside biology, with obviously limited biological understanding.**
>
> **Response**:
> We respectfully disagree with the reviewer’s claim that this work constitutes a “naive out-of-the-box application” of a non-biological method. In fact, substantial model-level and experiment-level adaptations were introduced to make BioSASANet biologically meaningful and genomics-specific. Our contributions go far beyond a direct reuse of an existing architecture:
>
> _**1. Genomics-specific model design:**_
>
> Our model design explicitly incorporates key characteristics of genomic sequences, including long-range nucleotide interactions and the reverse-complementary nature of DNA. Building upon the theoretically grounded self-interpretable SASANet framework, BioSASANet provides biologically meaningful nucleotide-level attributions for genomic sequences. These attribution scores offer strong potential for future high-stakes biological applications, such as variant effect interpretation and regulatory element discovery.
>
> **_2. Biologically grounded interpretability validation_**
>
> Unlike generic attribution methods, our evaluation focuses on biologically grounded correctness. Specifically: (1) **MP@k (Mean Precision at k)** measures whether high-scoring nucleotides align with biologically validated regions—core promoter element windows and histone-mark peak intervals—thus ensuring that the learned attributions correspond to known regulatory features. (2) **In silico mutagenesis** further tests attribution causality by quantifying the functional impact of perturbing each nucleotide, providing a direct measure of biological correctness beyond simple known motif overlap.
> Together, these biologically motivated validation strategies demonstrate substantial domain understanding, which goes beyond a “naive” application of non-biological methods.
>
> **_3. Adaptability across genomics backbones_**
>
> To prove BioSASANet's compatibility with different backbones for genomics sequential modeling, we further implement two variants of BioSASANet using different state-of-the-art backbone DNA LMs: **Evo2** and **HyenaDNA**. These two backbones represent distinct architectural families widely used in modern genomics modeling. BioSASANet integrates with both seamlessly, and consistently produces high-quality nucleotide-level attributions across them.
> Together, these demonstrate that the method is not merely transplanted from outside biology, but has been carefully adapted to encode biologically meaningful priors and evaluated through biologically motivated experiments.
>
> **Reviewer comment 2: Experiments compare against outdated methods on outdated data that is not at the single-nucleotide resolution that Shapley-value-based interpretation is supposed to be foucsed on.**
>
> **Response**:
>
> We respectfully disagree with the reviewer’s claim that our comparisons rely on outdated methods or data.
>
> **1. Baselines are not outdated.**
>
> The compared attribution methods (DeepLIFT, GradientSHAP, LIME) remain the *standard and most widely used* post-hoc interpretability baselines in genomics, adopted in recent works such as BPNet [1], DeepSTARR [2], and related regulatory-sequence studies [3]. In addition, we validate BioSASANet on two **state-of-the-art DNA language models**—**HyenaDNA** and **Evo2**—demonstrating compatibility with the latest backbone architectures.
>
> **2. Data are not outdated and are fully single-nucleotide resolution.**
>
> Both of our datasets provide **base-level ground-truth**, including:
>
> - **K562 histone-mark peaks (ENCODE)** — standard single-nucleotide–resolution epigenomic labels used in modern genomics models.
> - **Plant core promoter datasets** — experimentally validated CPE windows at single-base precision, used in recent *Nature Plants* paper.
>
> These datasets are **updated, biologically relevant**, and specifically suited for evaluating nucleotide-level Shapley-based attributions.
>
> **In summary**, our baselines, backbones, and datasets reflect *current* standards in regulatory genomics, and the comparisons are neither outdated nor misaligned with modern practice.
>
> [1] Avsec, Ž., Weilert, M., Shrikumar, A., Krueger, S., Alexandari, A., Dalal, K., ... & Zeitlinger, J. (2021). Base-resolution models of transcription-factor binding reveal soft motif syntax. *Nature genetics*, *53*(3), 354-366.
>
> [2] de Almeida, B. P., Reiter, F., Pagani, M., & Stark, A. (2022). DeepSTARR predicts enhancer activity from DNA sequence and enables the de novo design of synthetic enhancers. *Nature genetics*, *54*(5), 613-624.
>
> [3] Chen, V., Yang, M., Cui, W., Kim, J. S., Talwalkar, A., & Ma, J. (2024). Applying interpretable machine learning in computational biology—pitfalls, recommendations and opportunities for new developments. *Nature methods*, *21*(8), 1454-1461.

---

> > ### Comment · Reviewer_7y8v · 2025-11-26
> > **Response to rebuttal**
> >
> > Thanks authors for their response.
> >
> >
> > 1. Genomics-specific model design:
> > The reverse complementarity and long range effects are acknowledged, but are contributions of previous publications cited in this work. A responsive representation would elevate above the raw sequence level.
> >
> > 2.  Biologically grounded interpretability validation
> > Being part of a low-resolution ChIPSeq peak as a figure of merit (for precision for MP@k) is a demonstration of misunderstanding the biological phenomena of histone modification and its resolution. These modification are more localized than measured using ChIPSeq, so while the authors should be congratulated for capturing the artifacts of ChIPSeq technology better than methods from the mid 2010s, this does not address the biological understanding critique
> >
> >
> > 3. Adaptability across genomics backbones
> > A different way of saying this is that the authors’ chosen backbone is of limited contribution
> >
> > 4. Baselines outdated:
> > The comps DeepLift, GradientSHAP, and Lime were great when published (2016-7), but that’s eons ago in the AI world.
> >
> > 5. Data outdated:
> > The ENCODE data is ChIPSeq, state of the art from 2012, since superceded by Cut&Run. Cut&Tag technologies with higher nucleotide resolution. Even in the slow world of developing lab protocols, the data is outdated.
> >
> > 6. ISM:
> > The ground truth is not ground truth It is a different model.

---

> > > ### Author Response · Authors · 2025-12-01
> > > **Response to reviewer 7y8v's round 2 comments**
> > >
> > > ## Response to 1:
> > > We acknowledge that the reverse-complement and long-range modeling components follow prior work, since our focus is not on proposing new sequence-representation mechanisms, but on introducing a **theoretically grounded, self-interpretable, and adaptable framework** for nucleotide-level attribution. These prior modules are used only to obtain (i) **prefix representations $x_{O_{1:i-1}}$** for marginal contributions $\Delta(x_{O_i}, x_{O_{1:i-1}};\theta_\Delta)$ (NMCSM) and (ii) **context-aware representations $\mathbf{x}$** for positional Shapley values $\phi(\mathbf{x}; \theta_\phi)_{i, k}$  for feature i appearing at position k (PSVM).
> > >
> > > Regarding the reviewer’s suggestion that **“a responsive representation would elevate above the raw sequence level”**: this is indeed an interesting direction—for example, learning higher-order biologically meaningful representations beyond the nucleotide sequence. However, our goal in this paper is specifically to derive **nucleotide-level attributions**, which intrinsically requires operating at the raw-sequence level. We consider exploring higher-level representations as valuable future work but beyond the scope of the current study.
> > >
> > > ## Response to 2:
> > > We sincerely thank the reviewer for raising this important point regarding biological grounding. Our goal in this work is to develop a **self-interpretable neural network that produces faithful nucleotide-level attributions**, rather than to study histone biology itself. We therefore follow the same evaluation setup used in recent genomics foundation models—such as Nucleotide Transformer [1]—which formulate histone-mark prediction as a sequence-level binary classification task using ChIP-seq peak intervals inclusion as positive labels. Consistent with these prior works, we use peak annotations only as a **proxy** for assessing whether the learned nucleotide attributions align with known regulatory regions.
> > >
> > > We fully agree that histone modifications are more localized than what ChIP-seq resolutions capture, and MP@k based on peak intervals cannot perfectly reflect the true biochemical specificity. This is precisely why we additionally include **in-silico mutagenesis (ISM)** analyses, which provide a more mechanistic and fine-grained causal validation at single-nucleotide resolution.
> > >
> > > We appreciate the reviewer’s suggestion and would be grateful for recommendations of **higher-resolution or biologically richer benchmarks** for validating nucleotide-level attribution. Incorporating such datasets would be valuable future work.
> > > ## Response to 3:
> > > We thank the reviewer for this comment. Our goal is not to introduce a new model for sequence representation, but to propose a **theoretically grounded, self-interpretable, and adaptable attribution framework**. BioSASANet requires two types of representations for each nucleotide—**prefix representations** (for computing marginal contributions) and **context representations** (for positional Shapley values). These representations can be supplied by any causal (for prefix representation) or non-causal (for context representation) genomic sequential backbones.
> > >
> > > To demonstrate this flexibility, we instantiated BioSASANet with three different backbones—**Mamba**, **Evo2-7b**, and **HyenaDNA**—and showed that the framework functions consistently across all of them. This experiment evaluates and confirms the adaptability and modularity of BioSASANet, rather than the contribution of the backbones themselves.
> > >
> > >
> > >
> > >
> > > [1] Dalla-Torre, H., Gonzalez, L., Mendoza-Revilla, J., Lopez Carranza, N., Grzywaczewski, A. H., Oteri, F., ... & Pierrot, T. (2025). Nucleotide Transformer: building and evaluating robust foundation models for human genomics. *Nature Methods*, *22*(2), 287-297.

---

> > > ### Author Response · Authors · 2025-12-01
> > > **Response to reviewer 7y8v's round 2 comments: 4-6**
> > >
> > > ## Response to 4:
> > >
> > > We respectfully disagree with the reviewer’s claim that DeepLIFT, GradientSHAP, and LIME are “outdated”. While these methods were originally developed in 2016–2017, **they remain the dominant and most widely used attribution techniques in modern computational genomics**, as evidenced by multiple high-impact recent studies.
> > >
> > > - **BPNet [2]** (Avsec et al., *Nature Genetics 2021*) uses **DeepLIFT** for nucleotide-resolution TF-binding attribution.
> > > - **DeepSTARR [3]** (de Almeida et al., *Nature Genetics 2022*) uses **DeepLIFT** to characterize enhancer syntax.
> > > - **Enformer [4]** (Avsec, Ž.et al., *Nature Methods 2021*) uses **Gradient × Input** for interpreting long-range enhancer–promoter interactions.
> > > - **Borzoi [5]** (Linder et al., Nature Genetics 2025) uses **Gradient × Input** to derive nucleotide-level attributions and identify cis-regulatory motifs driving RNA expression.
> > >
> > > These works demonstrate that these attribution algorithms are **still standard, authoritative baselines** for biological interpretability. Furthermore, in response to the reviewer’s suggestion, **we have additionally included Grad × Input** and evaluated it across both datasets using MP@k, Spearman correlation with ISM, and overlap@k with ISM.
> > >
> > > Thus, our comparison covers:
> > >
> > > - **DeepLIFT**
> > > - **GradientSHAP**
> > > - **LIME**
> > > - **Grad × Input** (added per reviewer request)
> > > - **ISM-based causal evaluation** (gold-standard perturbation method)
> > >
> > > ## Response to 5:
> > >
> > > We appreciate the reviewer’s comment regarding data recency.
> > >
> > > (1) Although our histone-mark datasets are derived from ChIP-seq, this choice aligns with current high-impact genomics literature, the 2025 Nature Methods paper **“Nucleotide Transformer: building and evaluating robust foundation models for human genomics” [1]**.  They also adopt histone ChIP-seq peaks as a standard downstream benchmark. Thus, our data selection is consistent with prevailing practices in state-of-the-art computational genomics research.
> > >
> > > (2) We agree that newer high-resolution profiling technologies such as **CUT&RUN** and **CUT&Tag** offer improved signal quality and provide promising alternatives for future evaluation. We will incorporate these datasets in subsequent work to further assess BioSASANet’s nucleotide-level attribution accuracy under more modern experimental protocols.
> > >
> > > ## Response to 6:
> > > We respectfully clarify that we do not claim ISM provides biological ground-truth. Rather, following established practice in recent high-impact genomics work, we use ISM as a **proxy** for nucleotide-level causal effects. Multiple studies, for example Borzoi [5], explicitly validate model attributions by comparing them against ISM scores, noting that ISM yields more reliable functional perturbation signals than gradient-based methods in several regulatory contexts.
> > >
> > > Importantly, ISM is also used as a biologically grounded perturbation surrogate in experimental settings. For example, Karollus, A. et al.  [6] use ISM to replicate CRISPRi knockdown experiments by quantifying changes in predicted expression when disrupting enhancer sequences. This demonstrates that ISM is widely accepted as one faithful **single-nucleotide perturbation reference** available in silico.
> > >
> > > Therefore, our use of ISM, as a high-fidelity causal proxy to assess correlation/overlap with different attribution methods, is fully aligned with current standards in computational genomics.
> > >
> > >
> > >
> > > [1] Dalla-Torre, H., Gonzalez, L., Mendoza-Revilla, J., Lopez Carranza, N., Grzywaczewski, A. H., Oteri, F., ... & Pierrot, T. (2025). Nucleotide Transformer: building and evaluating robust foundation models for human genomics. *Nature Methods*, *22*(2), 287-297.
> > >
> > > [2] Avsec, Ž., Weilert, M., Shrikumar, A., Krueger, S., Alexandari, A., Dalal, K., ... & Zeitlinger, J. (2021). Base-resolution models of transcription-factor binding reveal soft motif syntax. *Nature genetics*, *53*(3), 354-366.
> > >
> > > [3] de Almeida, B. P., Reiter, F., Pagani, M., & Stark, A. (2022). DeepSTARR predicts enhancer activity from DNA sequence and enables the de novo design of synthetic enhancers. *Nature genetics*, *54*(5), 613-624.
> > >
> > > [4] Avsec, Ž., Agarwal, V., Visentin, D., Ledsam, J. R., Grabska-Barwinska, A., Taylor, K. R., ... & Kelley, D. R. (2021). Effective gene expression prediction from sequence by integrating long-range interactions. *Nature methods*, *18*(10), 1196-1203.
> > >
> > > [5] Linder, J., Srivastava, D., Yuan, H., Agarwal, V., & Kelley, D. R. (2025). Predicting RNA-seq coverage from DNA sequence as a unifying model of gene regulation. *Nature Genetics*, *57*(4), 949-961.
> > >
> > > [6] Karollus, A., Mauermeier, T., & Gagneur, J. (2023). Current sequence-based models capture gene expression determinants in promoters but mostly ignore distal enhancers. *Genome biology*, *24*(1), 56.

---

> ### Author Response · Authors · 2025-11-23
> **Responses for the questions.**
>
> **1. On "advanced feature-determinaiton methods on single-nucleotide-sensitive features":**
>
> To directly address the reviewer’s request, we have added **in silico mutagenesis (ISM)** experiments on both datasets. ISM is widely regarded as the gold-standard single-nucleotide perturbation method for validating causal feature importance in genomic models.
>
> For each test sequence, we compute ISM scores and then compare **BioSASANet** and three **post-hoc attribution methods** using: (1) Spearman correlation ρ, (2) Top-k overlap between the most influential positions from attribution scores and ISM scores. The added experiments' results are shown below:
>
> On plant datasets:
>
> | **Eval Metric**          | **Dataset** | **BioSASANet**      | **DeepLift**     | **GradientSHAP** | **Lime**         |
> | ------------------------ | ----------- | ------------------- | ---------------- | ---------------- | ---------------- |
> | **Spearman correlation** | Plant leaf  | **0.5022 ± 0.1145** | 0.0646 ± 0.1111  | 0.0024 ± 0.0974  | −0.0045 ± 0.0811 |
> |                          | Plant proto | **0.5439 ± 0.1310** | −0.0694 ± 0.1245 | −0.0611 ± 0.1494 | −0.0026 ± 0.0822 |
> | **Overlap@30**           | Plant leaf  | **0.4926**          | 0.3270           | 0.2624           | 0.1230           |
> |                          | Plant proto | **0.5846**          | 0.2681           | 0.2971           | 0.0863           |
> | **Overlap@50**           | Plant leaf  | **0.5703**          | 0.4003           | 0.3508           | 0.2222           |
> |                          | Plant proto | **0.6229**          | 0.3300           | 0.3391           | 0.1225           |
> | **Overlap@100**          | Plant leaf  | **0.7323**          | 0.5973           | 0.5879           | 0.6059           |
> |                          | Plant proto | **0.7329**          | 0.5579           | 0.5473           | 0.4512           |
>
> On histone mark datasets:
>
> | **Eval Metric**           | **Datasets** | **BioSASANet**      | **DeepLift**    | **GradientSHAP** | **Lime**        |
> | ------------------------- | ------------ | ------------------- | --------------- | ---------------- | --------------- |
> | **Spearman with ISM (ρ)** | H3K4me3      | **0.6307 ± 0.0169** | 0.1320 ± 0.0532 | 0.1114 ± 0.0412  | 0.0035 ± 0.0323 |
> |                           | H3K27me3     | **0.6462 ± 0.0163** | 0.0323 ± 0.0524 | 0.0259 ± 0.0502  | 0.0031 ± 0.0299 |
> |                           | H3K9me3      | **0.6218 ± 0.0157** | 0.0449 ± 0.0288 | 0.0515 ± 0.0278  | 0.0013 ± 0.0306 |
> |                           | H3K36me3     | **0.6793 ± 0.0159** | 0.0707 ± 0.0457 | 0.0539 ± 0.0363  | 0.0014 ± 0.0283 |
> | **Overlap@50**            | H3K4me3      | **0.5048**          | 0.2624          | 0.2309           | 0.0269          |
> |                           | H3K27me3     | **0.5234**          | 0.2927          | 0.2303           | 0.0300          |
> |                           | H3K9me3      | **0.4960**          | 0.2765          | 0.2403           | 0.0235          |
> |                           | H3K36me3     | **0.6087**          | 0.3741          | 0.2964           | 0.0109          |
> | **Overlap@100**           | H3K4me3      | **0.5502**          | 0.3229          | 0.2934           | 0.0684          |
> |                           | H3K27me3     | **0.5666**          | 0.3433          | 0.2774           | 0.0637          |
> |                           | H3K9me3      | **0.5511**          | 0.3119          | 0.2880           | 0.0473          |
> |                           | H3K36me3     | **0.6409**          | 0.3891          | 0.3043           | 0.0099          |
> | **Overlap@200**           | H3K4me3      | **0.6146**          | 0.3816          | 0.3559           | 0.1502          |
> |                           | H3K27me3     | **0.6392**          | 0.3854          | 0.3340           | 0.1404          |
> |                           | H3K9me3      | **0.6073**          | 0.3667          | 0.3531           | 0.1137          |
> |                           | H3K36me3     | **0.6968**          | 0.4210          | 0.3579           | 0.0484          |
>
> Across all datasets and evaluation metrics, BioSASANet shows the strongest agreement with ISM—significantly outperforming other baselines.
>
> **2. On "Figures of merit should involved clinical meaning or changes in downstream expression levels" :** We would like to clarify that our evaluations already incorporate biologically grounded signals. For **histone-mark data**, MP@k measures alignment between high-attribution positions and experimentally observed peak intervals (Table 1), and Figure 1 shows clear visual correspondence. For **plant promoters**, MP@k evaluates agreement with core promoter elements (e.g., TATA box, Y-patch) with functional relevance (Table 2), and Figure 3 in appendix further illustrates precise alignment in case studies. Together, these results provide functionally meaningful and biologically grounded validation, as requested.

---

### Author Response · Authors · 2025-12-04
**To Area Chair**

We sincerely thank the Area Chair for their time and thoughtful coordination throughout the review process.

Our paper introduces BioSASANet, a self-interpretable and architecture-agnostic genomic sequence model that provides theoretically grounded, nucleotide-level attributions while modeling key biological properties such as reverse complementarity and long-range dependencies.

Reviewers acknowledged several strengths of the work, including its clear motivation for biological interpretability, competitive attribution performance, and architectural flexibility. lmportantly, reviewers recognized our potential to provide biological insights.

In the rebuttal, we addressed all concerns by adding substantial new experiments and clarifications including **lSM-based causal validation**, **Backbone generalization**, **Module ablations**, **Efficiency analysis**, **improved case studies &formatting**.

We thank the reviewers for their constructive feedback, and we hope the revised manuscript clearly reflects the technical soundness, biological relevance, and potential impact of the proposed framework.

---

### Meta-Review · Area_Chair_jpJs · 2026-01-07

**Summary:**

This paper presents BioSASANet, a self-interpretable framework for genomic sequence modeling that adapts the SASA formulation to produce nucleotide-level attributions while supporting properties such as reverse complementarity and long-range dependencies. The topic is clearly important, and multiple reviewers agreed that intrinsic interpretability for genomics is a meaningful and worthwhile direction.

The rebuttal was **very strong**. The authors clearly invested substantial effort and responded carefully to nearly all reviewer concerns, adding in-silico mutagenesis experiments, additional attribution baselines, module ablations, efficiency analysis, and clarifications on backbone generalization and evaluation protocol. Compared to the initial submission, the revised version is significantly improved, and the technical soundness of the approach is much clearer.

That said, even after the rebuttal, it is difficult to say that there is sufficient alignment across reviewers to confidently recommend acceptance at this venue. In particular, concerns around the level of conceptual novelty relative to prior SASA-based work, as well as the biological grounding of the evaluation and validation choices, were not fully resolved to the point of consensus. While these issues do not imply that the work is incorrect, they do affect how the contribution is perceived at this stage.

Given the overall score distribution and the remaining disagreement in reviewer assessments, I ultimately lean toward a borderline reject. This is a close call. The paper is clearly on an upward trajectory, and the rebuttal meaningfully strengthened the work, but overturning the initial evaluations would be difficult to justify.

I want to emphasize that this decision should not be read as a negative assessment of the work. With tighter positioning of the core contribution, clearer separation from prior SASA frameworks, and possibly further biological validation or reframing, this paper would be very competitive at a future venue. I strongly encourage the authors to continue developing this line of work and to resubmit.

**Reviewer Concerns:**

- Reviewer 7y8v.

While the rebuttal added substantial new experiments (including ISM and additional baselines), the reviewer’s core concerns about biological grounding, data recency, and the appropriateness of the evaluation criteria were only partially addressed. This disagreement appears largely conceptual and remained unresolved.

- Reviewer cZLE.

The main concerns were largely addressed through added ISM validation, module ablations, backbone generalization, and efficiency analysis. Remaining issues are relatively minor and mostly related to evaluation breadth.

- Reviewer SobZ.

Most technical concerns were addressed, including additional backbone comparisons, ablations, efficiency measurements, and clearer case-study explanations. Some requests for broader evaluation remain, but the core criticisms were mitigated.

- Reviewer L7YJ.

Presentation issues, missing figures, and literature coverage were acknowledged and addressed. However, skepticism regarding the level of conceptual novelty and the limited evaluation scope was only partially resolved.

**Reviewer Scores:**

Reviewer 7y8v & cZLE would likely unchange their review scores.

Reviewer SobZ &  L7YJ could possible have a slight increase in their scores.

---

### Decision · Program_Chairs · 2026-01-26

Reject